# LOOKAHEAD: A FAR-SIGHTED ALTERNATIVE OF MAGNITUDE-BASED PRUNING

**Sejun Park**[*][†]**, Jaeho Lee**[*][†‡]**, Sangwoo Mo**[†] **and Jinwoo Shin**[†‡]
[†] KAIST EE    [‡] KAIST AI
{sejun.park,jaeho-lee,swmo,jinwoos}@kaist.ac.kr

## ABSTRACT

Magnitude-based pruning is one of the simplest methods for pruning neural networks. Despite its simplicity, magnitude-based pruning and its variants demonstrated remarkable performances for pruning modern architectures. Based on the observation that magnitude-based pruning indeed minimizes the Frobenius distortion of a linear operator corresponding to a single layer, we develop a simple pruning method, coined lookahead pruning, by extending the single layer optimization to a multi-layer optimization. Our experimental results demonstrate that the proposed method consistently outperforms magnitude-based pruning on various networks, including VGG and ResNet, particularly in the high-sparsity regime. See https://github.com/alinlab/lookahead_pruning for codes.

## 1 INTRODUCTION

The "magnitude-equals-saliency" approach has been long underlooked as an overly simplistic baseline among all imaginable techniques to eliminate unnecessary weights from over-parametrized neural networks. Since the early works of LeCun et al. (1989); Hassibi & Stork (1993) which provided more theoretically grounded alternatives of magnitude-based pruning (MP) based on second derivatives of the loss function, a wide range of methods including Bayesian / information-theoretic approaches (Neal, 1996; Louizos et al., 2017; Molchanov et al., 2017; Dai et al., 2018), $\ell_p$-regularization (Wen et al., 2016; Liu et al., 2017; Louizos et al., 2018), sharing redundant channels (Zhang et al., 2018; Ding et al., 2019), and reinforcement learning approaches (Lin et al., 2017; Bellec et al., 2018; He et al., 2018) have been proposed as more sophisticated alternatives.

On the other hand, the capabilities of MP heuristics are gaining attention once more. Combined with minimalistic techniques including iterative pruning (Han et al., 2015) and dynamic reestablishment of connections (Zhu & Gupta, 2017), a recent large-scale study by Gale et al. (2019) claims that MP can achieve a state-of-the-art trade-off between sparsity and accuracy on ResNet-50. The unreasonable effectiveness of magnitude scores often extends beyond the strict domain of network pruning; a recent experiment by Frankle & Carbin (2019) suggests the existence of an automatic subnetwork discovery mechanism underlying the standard gradient-based optimization procedures of deep, over-parametrized neural networks by showing that the MP algorithm finds an efficient trainable subnetwork. These observations constitute a call to revisit the "magnitude-equals-saliency" approach for a better understanding of the deep neural network itself.

As an attempt to better understand the nature of MP methods, we study a generalization of magnitude scores under a *functional approximation* framework; by viewing MP as a relaxed minimization of distortion in layerwise operators introduced by zeroing out parameters, we consider a multi-layer extension of the distortion minimization problem. Minimization of the newly suggested distortion measure, which 'looks ahead' the impact of pruning on neighboring layers, gives birth to a novel pruning strategy, coined *lookahead pruning* (LAP).

In this paper, we focus on the comparison of the proposed LAP scheme to its MP counterpart. We empirically demonstrate that LAP consistently outperforms MP under various setups, including linear networks, fully-connected networks, and deep convolutional and residual networks. In particular, LAP consistently enables more than $\times 2$ gain in the compression rate of the considered models, with

---

[*]equal contribution

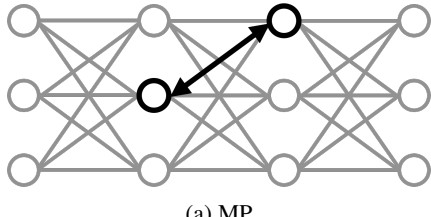 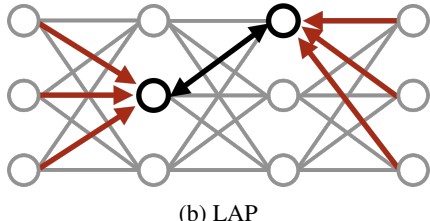

(a) MP                    (b) LAP

Figure 1: An illustration of magnitude-based pruning (MP) and lookahead pruning (LAP). MP only considers a single weight while LAP also considers the effects of neighboring edges.

increasing benefits under the high-sparsity regime. Apart from its performance, lookahead pruning enjoys additional attractive properties:

- *Easy-to-use:* Like magnitude-based pruning, the proposed LAP is a simple score-based approach agnostic to model and data, which can be implemented by computationally light elementary tensor operations. Unlike most Hessian-based methods, LAP does not rely on the availability of training data except for the retraining phase. It also has no hyper-parameter to tune, in contrast to other sophisticated training-based and optimization-based schemes.

- *Versatility:* As our method simply replaces the "magnitude-as-saliency" criterion with a lookahead alternative, it can be deployed jointly with algorithmic tweaks developed for magnitude-based pruning, such as iterative pruning and retraining (Han et al., 2015) or joint pruning and training with dynamic reconnections (Zhu & Gupta, 2017; Gale et al., 2019).

The remainder of this manuscript is structured as follows: In Section 2, we introduce a functional approximation perspective toward MP and motivate LAP and its variants as a generalization of MP for multiple layer setups; in Section 3 we explore the capabilities of LAP and its variants with simple models, then move on to apply LAP to larger-scale models.

## 2 LOOKAHEAD: A FAR-SIGHTED LAYER APPROXIMATION

We begin by a more formal description of the magnitude-based pruning (MP) algorithm (Han et al., 2015). Given an $L$-layer neural network associated with weight tensors $W_1, \ldots, W_L$, the MP algorithm removes connections with the smallest absolute weights from each weight tensor until the desired level of sparsity has been achieved. This layerwise procedure is equivalent to finding a mask $M$ whose entries are either 0 or 1, incurring a smallest *Frobenius distortion*, measured by

$$\min_{M:\|M\|_0=s} \|W - M \odot W\|_F, \tag{1}$$

where $\odot$ denotes the Hadamard product, $\|\cdot\|_0$ denotes the entrywise $\ell_0$-norm, and $s$ is a sparsity constraint imposed by some operational criteria.

Aiming to minimize the Frobenius distortion (Eq. (1)), the MP algorithm naturally admits a functional approximation interpretation. For the case of a fully-connected layer, the maximal difference between the output from a pruned and an unpruned layer can be bounded as

$$\|Wx - (M \odot W)x\|_2 \leq \|W - M \odot W\|_2 \cdot \|x\|_2 \leq \|W - M \odot W\|_F \cdot \|x\|_2. \tag{2}$$

Namely, the product of the layerwise Frobenius distortion upper bounds the output distortion of the network incurred by pruning weights. Note that this perspective on MP as a worst-case distortion minimization was already made in Dong et al. (2017), which inspired an advent of the layerwise optimal brain surgery (L-OBS) procedure.

A similar idea holds for convolutional layers. For the case of a two-dimensional convolution with a single input and a single output channel, the corresponding linear operator takes a form of a doubly block circulant matrix constructed from the associated kernel tensor (see, e.g., Goodfellow et al. (2016)). Here, the Frobenius distortion of doubly block circulant matrices can be controlled by the Frobenius distortion of the weight tensor of the convolutional layer.[1]

---

[1]The case of multiple input/output channels or non-circular convolution can be dealt with similarly using channel-wise circulant matrices as a block. We refer the interested readers to Sedghi et al. (2019).

---

**Algorithm 1** Lookahead Pruning (LAP)

---

1: **Input:** Weight tensors $W_1, \ldots, W_L$ of a trained network, desired sparsities $s_1, \ldots, s_L$
2: **Output:** Pruned weight tensors $\widetilde{W}_1, \ldots, \widetilde{W}_L$
3: **for** $i = 1, \ldots, L$ **do**
4: $\quad$ Compute $\mathcal{L}_i(w)$ according to Eq. (4) for all entry $w$ of $W_i$
5: $\quad$ Set $\tilde{w}_{s_i}$ as a $s_i$-th smallest element of $\{\mathcal{L}_i(w) : w \text{ is an entry of } W_i\}$
6: $\quad$ Set $M_i \leftarrow \mathbb{1}\{W_i - \tilde{w}_{s_i} \geq 0\}$
7: $\quad$ Set $\widetilde{W}_i \leftarrow M_i \odot W_i$
8: **end for**

---

## 2.1 LOOKAHEAD DISTORTION AS A BLOCK APPROXIMATION ERROR

The myopic optimization (Eq. (1)) based on the per-layer Frobenius distortion falls short even in the simplest case of the two-layer linear neural network with one-dimensional output, where we consider predictors taking form $\widehat{Y} = u^\top W x$ and try to minimize the Frobenius distortion of $u^\top W$ (equivalent to $\ell_2$ distortion in this case). Here, if $u_i$ is extremely large, pruning any nonzero element in the $i$-th row of $W$ may incur a significant Frobenius distortion.

Motivated by this observation, we consider a *block approximation* analogue of the magnitude-based pruning objective Eq. (1). Consider an $L$-layer neural network associated with weight tensors $W_1, \ldots, W_L$, and assume linear activation for simplicity (will be extended to nonlinear cases later in this section). Let $\mathcal{J}(W_i)$ denote the Jacobian matrix corresponding to the linear operator characterized by $W_i$. For pruning the $i$-th layer, we take into account the weight tensors of adjacent layers $W_{i-1}, W_{i+1}$ in addition to the original weight tensor $W_i$. In particular, we propose to minimize the Frobenius distortion of the operator block $\mathcal{J}(W_{i+1})\mathcal{J}(W_i)\mathcal{J}(W_{i-1})$, i.e.,

$$\min_{M_i : \|M_i\|_0 = s_i} \|\mathcal{J}(W_{i+1})\mathcal{J}(W_i)\mathcal{J}(W_{i-1}) - \mathcal{J}(W_{i+1})\mathcal{J}(M_i \odot W_i)\mathcal{J}(W_{i-1})\|_F. \tag{3}$$

An explicit minimization of the block distortion (Eq. (3)), however, is computationally intractable in general (see Appendix D for a more detailed discussion).

To avoid an excessive computational overhead, we propose to use the following score-based pruning algorithm, coined lookahead pruning (LAP), for approximating Eq. (3): For each tensor $W_i$, we prune the weights $w$ with the smallest value of lookahead distortion (in a single step), defined as

$$\mathcal{L}_i(w) := \|\mathcal{J}(W_{i+1})\mathcal{J}(W_i)\mathcal{J}(W_{i-1}) - \mathcal{J}(W_{i+1})\mathcal{J}(W_i|_{w=0})\mathcal{J}(W_{i-1})\|_F \tag{4}$$

where $W_i|_{w=0}$ denotes the tensor whose entries are equal to the entries of $W_i$ except for having zeroed out $w$. We let both $W_0$ and $W_{L+1}$ to be tensors consisting of ones. In other words, lookahead distortion (Eq. (4)) measures the distortion (in Frobenius norm) induced by pruning $w$ while all other weights remain intact. For three-layer blocks consisting only of fully-connected layers and convolutional layers, Eq. (4) reduces to the following compact formula: for an edge $w$ connected to the $j$-th input neuron/channel and the $k$-th output neuron/channel of the $i$-th layer, where its formal derivation is presented in Appendix E.

$$\mathcal{L}_i(w) = |w| \cdot \left\|W_{i-1}[j,:]\right\|_F \cdot \left\|W_{i+1}[:,k]\right\|_F, \tag{5}$$

where $|w|$ denotes the weight of $w$, $W[j,:]$ denotes the slice of $W$ composed of weights connected to the $j$-th output neuron/channel, and $W[:,k]$ denotes the same for the $k$-th input neuron/channel.

In LAP, we compute the lookahead distortion for all weights, and then remove weights with the smallest distortions in a single step (as done in MP). A formal description of LAP is presented in Algorithm 1. We also note the running time of LAP is comparable with that of MP (see Appendix G).

**LAP on linear networks.** To illustrate the benefit of lookahead, we evaluate the performance of MP and LAP on a linear fully-connected network with a single hidden layer of 1,000 nodes, trained with the MNIST image classification dataset. Fig. 2a and Fig. 2b depict the test accuracy of models pruned with each method, before and after retraining steps.

As can be expected from the discrepancy between the minimization objectives (Eqs. (1) and (3)), networks pruned with LAP outperform networks pruned with MP at every sparsity level, in terms

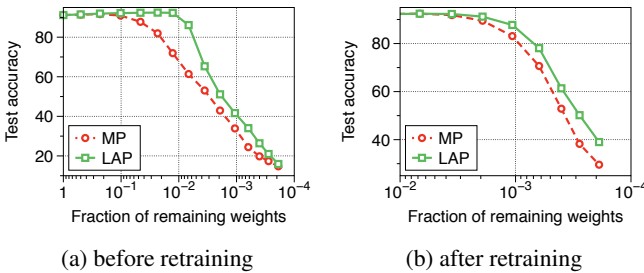

Figure 2: Test accuracy of pruned linear network under varying levels of sparsity, (a) before and (b) after a retraining phase. MP denotes magnitude-based pruning and LAP denotes lookahead pruning. All reported points are averaged over 5 trials.

of its performance before a retraining phase. Remarkably, we observe that test accuracy of models pruned with LAP monotonically increases from 91.2% to 92.3% as the sparsity level increases, until the fraction of surviving weights reaches 1.28%. At the same sparsity level, models pruned with MP achieves only 71.9% test accuracy. We also observe that LAP leads MP at every sparsity level even after a retraining phase, with an increasing margin as we consider a higher level of sparsity.

**Understanding LAP with nonlinear activations.** Most neural network models in practice deploy nonlinear activation functions, e.g., rectified linear units (ReLU). Although the lookahead distortion has been initially derived using linear activation functions, LAP can also be used for nonlinear networks, as the quantity $\mathcal{L}_i(w)$ remains relevant to the original block approximation point of view. This is especially true when the network is severely over-parametrized. To see this, consider a case where one aims to prune a connection in the first layer of a two-layer fully-connected network with ReLU, i.e.,

$$x \mapsto W_2\sigma(W_1 x), \tag{6}$$

where $\sigma(x) = \max\{0, x\}$ is applied entrywise. Under the over-parametrized scenario, zeroing out a single weight may alter the activation pattern of connected neurons with only negligible probability, which allows one to decouple the probability of activation of each neuron from the act of pruning each connection. This enables us to approximate the root mean square distortion of the network output introduced by pruning $w$ of $W_1$ by $\sqrt{p_k}\mathcal{L}_1(w)$, where $k$ is the index of the output neuron that $w$ is connected to, and $p_k$ denotes the probability of activation for the $k$-th neuron. In this sense, LAP (Algorithm 1) can be understood as assuming i.i.d. activations of neurons, due to a lack of additional access to training data. In other words, LAP admits a natural extension to the regime where we assume additional access to training data during the pruning phase. This variant, coined LAP-act, will be formally described in Appendix F, with experimental comparisons to another data-dependent baseline of optimal brain damage (OBD) (LeCun et al., 1989).

Another theoretical justification of using the lookahead distortion (Eq. (5)) for neural networks with nonlinear activation functions comes from recent discoveries regarding the implicit bias imposed by training via stochastic gradient descent (Du et al., 2018). See Appendix M for a detailed discussion.

As will be empirically shown in Section 3.1, LAP is an effective pruning strategy for sigmoids and tanh activations, that are not piece-wise linear as ReLU.

## 2.2 LOOKAHEAD PRUNING WITH BATCH NORMALIZATION

Batch normalization (BN), introduced by Ioffe & Szegedy (2015), aims to normalize the output of a layer per batch by scaling and shifting the outputs with trainable parameters. Based on our functional approximation perspective, having batch normalization layers in a neural network is not an issue for MP, which relies on the magnitudes of weights; batch normalization only affects the distribution of the input for each layer, not the layer itself. On the other hand, as the lookahead distortion (Eq. (3)) characterizes the distortion of the multi-layer block, one must take into account batch normalization when assessing the abstract importance of each connection.

The revision of lookahead pruning under the presence of batch normalization can be done fairly simply. Note that such a normalization process can be expressed as

$$x \mapsto a \odot x + b, \tag{7}$$

for some $a, b \in \mathbb{R}^{\dim(x)}$. Hence, we revise lookahead pruning to prune the connections with a minimum value of

$$\mathcal{L}_i(w) = |w| \cdot a_{i-1}[j] a_i[k] \cdot \left\| W_{i-1}[j, :] \right\|_F \cdot \left\| W_{i+1}[:, k] \right\|_F, \tag{8}$$

where $a_i[k]$ denotes the $k$-th index scaling factor for the BN layer placed at the output of the $i$-th fully-connected or convolutional layer (if BN layer does not exist, let $a_i[k] = 1$). This modification of LAP makes it an efficient pruning strategy, as will be empirically verified in Section 3.3.

## 2.3 VARIANTS OF LOOKAHEAD PRUNING

As the LAP algorithm (Algorithm 1) takes into account current states of the neighboring layers, LAP admits several variants in terms of lookahead direction, the order of pruning, and sequential pruning methods; these methods are extensively studied in Section 3.2. Along with "vanilla" LAP, we consider in total, six variants, which we now describe below:

**Mono-directional LAPs.**  To prune a layer, LAP considers both preceding and succeeding layers. Looking forward, i.e., only considering the succeeding layer, can be viewed as an educated modification of the internal representation the present layer produces. Looking backward, on the other hand, can be interpreted as only taking into account the expected structure of input coming into the current layer. The corresponding variants, coined LFP and LBP, are tested.

**Order of pruning.**  Instead of using the unpruned tensors of preceding/succeeding layers, we also consider performing LAP based on already-pruned layers. This observation brings up a question of the order of pruning; an option is to prune in a forward direction, i.e., prune the preceding layer first and use the pruned weight to prune the succeeding, and the other is to prune backward. Both methods are tested, which are referred to as LAP-forward and LAP-backward, respectively.

**Sequential pruning.**  We also consider a sequential version of LAP-forward/backward methods. More specifically, if we aim to prune total $p\%$ of weights from each layer, we divide the pruning budget into five pruning steps and gradually prune $(p/5)\%$ of the weights per step in forward/backward direction. Sequential variants will be marked with a suffix "-seq".

## 3 EXPERIMENTS

In this section, we compare the empirical performance of LAP with that of MP. More specifically, we validate the applicability of LAP to nonlinear activation functions in Section 3.1. In Section 3.2, we test LAP variants from Section 2.3. In Section 3.3, we test LAP on VGG (Simonyan & Zisserman, 2015), ResNet (He et al., 2016), and Wide ResNet (WRN, Zagoruyko & Komodakis (2016)).

**Experiment setup.**  We consider five neural network architectures: (1) The fully-connected network (FCN) under consideration is consist of four hidden layers, each with 500 neurons. (2) The convolutional network (Conv-6) consists of six convolutional layers, followed by a fully-connected classifier with two hidden layers with 256 neurons each; this model is identical to that appearing in the work of Frankle & Carbin (2019) suggested as a scaled-down variant of VGG.[2] (3) VGG-19 is used, with an addition of batch normalization layers after each convolutional layers, and a reduced number of fully-connected layers from three to one.[3] (4) ResNets of depths $\{18, 50\}$ are used. (5) WRN of 16 convolutional layers and widening factor 8 (WRN-16-8) is used. All networks used ReLU activation function, except for the experiments in Section 3.1. We mainly consider image classification tasks. In particular, FCN is trained on MNIST dataset (Lecun et al., 1998), Conv-6, VGG, and ResNet are trained on CIFAR-10 dataset (Krizhevsky & Hinton, 2009), and VGG, ResNet, and WRN are trained on Tiny-ImageNet.[4] We focus on the one-shot pruning of MP and LAP, i.e., models are trained with a single training-pruning-retraining cycle. All results in this section are averaged over five independent trials. We provide more details on setups in Appendix A.

---

[2]Convolutional layers are organized as $[64, 64] - \mathsf{MaxPool} - [128, 128] - \mathsf{MaxPool} - [256, 256]$.

[3]This is a popular configuration of VGG for CIFAR-10 (Liu et al., 2019; Frankle & Carbin, 2019)

[4]Tiny-ImageNet visual recognition challenge, https://tiny-imagenet.herokuapp.com.

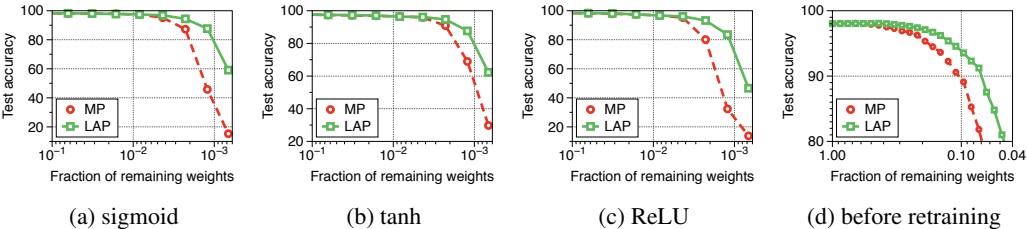

Figure 3: Test accuracy of FCN with (a) sigmoid, (b) tanh, (c) ReLU activations; (d) test accuracy of FCN with ReLU activation before retraining, for the MNIST dataset.

### 3.1 NETWORKS WITH NONLINEAR ACTIVATION FUNCTIONS

We first compare the performance of LAP with that of MP on FCN using three different types of activation functions: sigmoid, and tanh, and ReLU. Figs. 3a to 3c depict the performance of models pruned with LAP (Green) and MP (Red) under various levels of sparsity.

Although LAP was motivated primarily from linear networks and partially justified for positive-homogenous activation functions such as ReLU, the experimental results show that LAP consistently outperforms MP even on networks using sigmoidal activation functions. We remark that LAP outperforms MP by a larger margin as fewer weights survive (less than 1%). Such a pattern will be observed repeatedly in the remaining experiments of this paper.

In addition, we also check whether LAP still exhibits better test accuracy before retraining under the usage of nonlinear activation functions, as in the linear network case (Fig. 2b). Fig. 3d illustrates the test accuracy of pruned FCN using ReLU on the MNIST dataset before retraining. We observe that the network pruned by LAP continues to perform better than MP in this case; the network pruned by LAP retains the original test accuracy until only 38% of the weights survive, and shows less than 1% performance drop with only 20% of the weights remaining. On the other hand, MP requires 54% and 30% to achieve the same level of performance, respectively. In other words, the models pruned with MP requires about 50% more survived parameters than the models pruned with LAP to achieve a similar level of performance before being retrained using additional training batches.

### 3.2 EVALUATING LAP VARIANTS

Now we evaluate LAP and its variants introduced in Section 2.3 on FCN and Conv-6, each trained on MNIST and CIFAR-10, respectively. Table 1 summarizes the experimental results on FCN and Table 2 summarizes the results on Conv-6. In addition to the baseline comparison with MP, we also compare with random pruning (RP), where the connection to be pruned was decided completely independently. We observe that LAP performs consistently better than MP and RP with similar or smaller variance in any case. In the case of an extreme sparsity, LAP enjoys a significant performance gain; over 75% gain on FCN and 14% on Conv-6. This performance gain comes from a better training accuracy, instead of a better generalization; see Appendix L for more information.

Comparing mono-directional lookahead variants, we observe that LFP performs better than LBP in the low-sparsity regime, while LBP performs better in the high-sparsity regime; in any case, LAP performed better than both methods. Intriguingly, the same pattern appeared in the case of the ordered pruning. Here, LAP-forward can be considered an analogue of LBP in the sense that they both consider layers closer to the input to be more critical. Likewise, LAP-backward can be considered an analogue of LFP. We observe that LAP-forward performs better than LAP-backward in the high-sparsity regime, and vice versa in the low-sparsity regime. Our interpretation is as follows: Whenever the sparsity level is low, carefully curating the input signal is not important due to high redundancies in the natural image signal. This causes a relatively low margin of increment by looking backward in comparison to looking forward. When the sparsity level is high, the input signal is scarce, and the relative importance of preserving the input signal is higher.

Finally, we observe that employing forward/backward ordering and sequential methods leads to better performance, especially in the high-sparsity regime. There is no clear benefit of adopting directional methods in the low-sparsity regime. The relative gain in performance with respect to LAP is either marginal or unreliable.

Table 1: Test error rates of FCN on MNIST. Subscripts denote standard deviations, and bracketed numbers denote relative gains with respect to MP. Unpruned models have $1.98\%$ error rate.

|  | 6.36% | 3.21% | 1.63% | 0.84% | 0.43% | 0.23% | 0.12% |
|---|---|---|---|---|---|---|---|
| MP (baseline) | $1.75_{\pm0.11}$ | $2.11_{\pm0.14}$ | $2.53_{\pm0.09}$ | $3.32_{\pm0.27}$ | $4.77_{\pm0.22}$ | $19.85_{\pm8.67}$ | $67.62_{\pm9.91}$ |
| RP | $2.36_{\pm0.13}$ | $2.72_{\pm0.16}$ | $3.64_{\pm0.17}$ | $17.54_{\pm7.07}$ | $82.48_{\pm4.03}$ | $88.65_{\pm0.00}$ | $88.65_{\pm0.00}$ |
| LFP | $1.63_{\pm0.08}$ | $1.89_{\pm0.11}$ | $2.43_{\pm0.10}$ | $3.32_{\pm0.13}$ | $4.23_{\pm0.38}$ | $9.59_{\pm1.70}$ | $50.11_{\pm12.99}$ |
|  | (-6.41%) | (-10.60%) | (-3.95%) | (-0.12%) | (-11.40%) | (-51.70%) | (-25.91%) |
| LBP | $1.75_{\pm0.17}$ | $2.04_{\pm0.12}$ | $2.61_{\pm0.15}$ | $3.62_{\pm0.17}$ | $4.19_{\pm0.31}$ | $9.09_{\pm1.41}$ | $28.51_{\pm14.85}$ |
|  | (+0.69%) | (-3.31%) | (+3.00%) | (+8.97%) | (-12.23%) | (-54.21%) | (-57.84%) |
| LAP | $1.67_{\pm0.11}$ | $1.89_{\pm0.12}$ | $2.48_{\pm0.13}$ | $3.29_{\pm0.06}$ | $3.93_{\pm0.26}$ | $6.72_{\pm0.44}$ | $16.45_{\pm5.61}$ |
|  | (-4.24%) | (-10.61%) | (-2.05%) | (-1.08%) | (-17.72%) | (-66.15%) | (-75.68%) |
| LAP-forward | $1.60_{\pm0.08}$ | $1.93_{\pm0.15}$ | $2.51_{\pm0.11}$ | $3.56_{\pm0.19}$ | $4.47_{\pm0.20}$ | $6.58_{\pm0.33}$ | $12.00_{\pm0.73}$ |
|  | (-8.25%) | (-8.43%) | (-0.95%) | (+7.03%) | (-6.41%) | (-66.81%) | (-82.26%) |
| LAP-backward | $1.63_{\pm0.11}$ | $1.88_{\pm0.07}$ | $2.35_{\pm0.02}$ | $3.12_{\pm0.08}$ | $3.87_{\pm0.18}$ | $5.62_{\pm0.17}$ | $13.00_{\pm3.30}$ |
|  | (-6.64%) | (-10.80%) | (-7.03%) | (-6.08%) | (-19.02%) | (-71.71%) | (-80.78%) |
| LAP-forward-seq | $1.68_{\pm0.11}$ | $1.92_{\pm0.10}$ | $2.49_{\pm0.14}$ | $3.39_{\pm0.24}$ | $4.21_{\pm0.06}$ | $6.20_{\pm0.32}$ | $\mathbf{10.98_{\pm1.03}}$ |
|  | (-3.66%) | (-9.09%) | (-1.42%) | (+1.93%) | (-11.86%) | (-68.73%) | **(-83.76%)** |
| LAP-backward-seq | $\mathbf{1.57_{\pm0.08}}$ | $\mathbf{1.84_{\pm0.10}}$ | $\mathbf{2.20_{\pm0.10}}$ | $\mathbf{3.13_{\pm0.16}}$ | $\mathbf{3.62_{\pm0.14}}$ | $\mathbf{5.42_{\pm0.27}}$ | $11.92_{\pm4.61}$ |
|  | **(-10.08%)** | **(-12.41%)** | **(-13.27%)** | **(-5.90%)** | **(-24.13%)** | **(-72.71%)** | (-82.36%) |

Table 2: Test error rates of Conv-6 on CIFAR-10. Subscripts denote standard deviations, and bracketed numbers denote relative gains with respect to MP. Unpruned models have $11.97\%$ error rate.

|  | 10.62% | 8.86% | 7.39% | 6.18% | 5.17% | 4.32% | 3.62% |
|---|---|---|---|---|---|---|---|
| MP (baseline) | $11.86_{\pm0.33}$ | $12.20_{\pm0.21}$ | $13.30_{\pm0.30}$ | $15.81_{\pm0.59}$ | $20.19_{\pm2.35}$ | $24.43_{\pm1.48}$ | $28.60_{\pm2.10}$ |
| RP | $26.85_{\pm1.23}$ | $29.72_{\pm1.13}$ | $32.98_{\pm1.10}$ | $35.92_{\pm1.08}$ | $39.13_{\pm1.05}$ | $41.20_{\pm1.19}$ | $43.60_{\pm0.82}$ |
| LFP | $11.81_{\pm0.35}$ | $12.18_{\pm0.23}$ | $13.27_{\pm0.44}$ | $15.04_{\pm0.43}$ | $18.50_{\pm0.80}$ | $22.86_{\pm1.66}$ | $26.65_{\pm1.33}$ |
|  | (-0.39%) | (-0.20%) | (-0.26%) | (-4.87%) | (-8.37%) | (-6.40%) | (-6.83%) |
| LBP | $12.08_{\pm0.17}$ | $12.34_{\pm0.36}$ | $13.26_{\pm0.16}$ | $14.93_{\pm0.85}$ | $18.11_{\pm1.27}$ | $22.57_{\pm0.94}$ | $26.34_{\pm1.60}$ |
|  | (+1.84%) | (-1.15%) | (-0.33%) | (-5.57%) | (-10.31%) | (-7.59%) | (-7.91%) |
| LAP | $\mathbf{11.76_{\pm0.24}}$ | $\mathbf{12.16_{\pm0.27}}$ | $\mathbf{13.05_{\pm0.14}}$ | $14.39_{\pm0.44}$ | $17.10_{\pm1.26}$ | $21.24_{\pm1.16}$ | $24.52_{\pm1.11}$ |
|  | **(-0.83%)** | **(-0.34%)** | **(-1.86%)** | (-8.99%) | (-15.30%) | (-13.04%) | (-14.29%) |
| LAP-forward | $11.82_{\pm0.16}$ | $12.35_{\pm0.34}$ | $13.09_{\pm0.36}$ | $14.42_{\pm0.45}$ | $17.05_{\pm1.30}$ | $20.28_{\pm1.40}$ | $22.80_{\pm0.51}$ |
|  | (-0.33%) | (+1.24%) | (-1.62%) | (-8.79%) | (-15.57%) | (-16.98%) | (-20.30%) |
| LAP-backward | $11.82_{\pm0.25}$ | $12.29_{\pm0.06}$ | $12.93_{\pm0.38}$ | $14.55_{\pm0.58}$ | $17.00_{\pm0.84}$ | $20.00_{\pm0.82}$ | $23.37_{\pm1.16}$ |
|  | (-0.32%) | (+0.68%) | (-2.78%) | (-7.98%) | (-15.78%) | (-18.11%) | (-18.30%) |
| LAP-forward-seq | $12.01_{\pm0.17}$ | $12.47_{\pm0.37}$ | $13.19_{\pm0.19}$ | $\mathbf{14.12_{\pm0.28}}$ | $\mathbf{16.73_{\pm0.95}}$ | $\mathbf{19.63_{\pm1.81}}$ | $\mathbf{22.44_{\pm1.31}}$ |
|  | (+1.28%) | (+2.21%) | (-0.81%) | **(-10.70%)** | **(-17.13%)** | **(-19.62%)** | **(-21.54%)** |
| LAP-backward-seq | $11.81_{\pm0.16}$ | $12.35_{\pm0.26}$ | $13.25_{\pm0.21}$ | $14.17_{\pm0.44}$ | $16.99_{\pm0.97}$ | $19.94_{\pm1.02}$ | $23.15_{\pm1.12}$ |
|  | (-0.39%) | (+1.25%) | (-0.41%) | (-10.37%) | (-15.87%) | (-18.38%) | (-19.08%) |

## 3.3 DEEPER NETWORKS: VGG, RESNET, AND WRN

We also compare empirical performances of MP with LAP on deeper networks. We trained VGG-19 and ResNet-18 on CIFAR-10 (Tables 3 and 4), and VGG-19, ResNet-50, and WRN-16-8 on Tiny-ImageNet (Tables 5 to 7). For models trained on CIFAR-10, we also test LAP-forward to verify the observation that it outperforms LAP in the high-sparsity regime on such deeper models. We also report additional experimental results on VGG-$\{11, 16\}$ trained on CIFAR-10 in Appendix B. For models trained on Tiny-ImageNet, top-1 error rates are reported in Appendix C.

From Tables 3 to 7, we make the following two observations: First, as in Section 3.2, the models pruned with LAP consistently achieve a higher or similar level of accuracy compared to models pruned with MP, at all sparsity levels. In particular, test accuracies tend to decay at a much slower rate with LAP. In Table 3, for instance, we observe that the models pruned by LAP retain test accuracies of 70~80% even with less than 2% of weights remaining. In contrast, the performance of models pruned with MP falls drastically, to below 30% accuracy. This observation is consistent on both CIFAR-10 and Tiny-ImageNet datasets.

Second, the advantages of considering an ordered pruning method (LAP-forward) over LAP is limited. While we observe from Table 3 that LAP-forward outperforms both MP and LAP in the high-sparsity regime, the gain is marginal considering standard deviations. LAP-forward is consistently worse than LAP (by at most 1% in absolute scale) in the low-sparsity regime.

Table 3: Test error rates of VGG-19 on CIFAR-10. Subscripts denote standard deviations, and bracketed numbers denote relative gains with respect to MP. Unpruned models have 9.02% error rate.

| | 12.09% | 8.74% | 6.31% | 4.56% | 3.30% | 2.38% | 1.72% | 1.24% |
|---|---|---|---|---|---|---|---|---|
| MP (baseline) | $8.99_{\pm 0.12}$ | $9.90_{\pm 0.09}$ | $11.43_{\pm 0.24}$ | $15.62_{\pm 1.68}$ | $29.10_{\pm 8.78}$ | $40.27_{\pm 11.51}$ | $63.27_{\pm 11.91}$ | $77.90_{\pm 7.94}$ |
| LAP | $\mathbf{8.89}_{\pm \mathbf{0.14}}$ | $\mathbf{9.51}_{\pm \mathbf{0.22}}$ | $\mathbf{10.56}_{\pm \mathbf{0.28}}$ | $\mathbf{12.11}_{\pm \mathbf{0.44}}$ | $13.64_{\pm 0.77}$ | $16.38_{\pm 1.47}$ | $20.88_{\pm 1.71}$ | $22.82_{\pm 0.81}$ |
| | **(-1.07%)** | **(-3.96%)** | **(-7.63%)** | **(-22.48%)** | (-53.13%) | (-59.31%) | (-67.00%) | (-70.71%) |
| LAP-forward | $9.63_{\pm 0.25}$ | $10.31_{\pm 0.23}$ | $11.10_{\pm 0.22}$ | $12.24_{\pm 0.33}$ | $\mathbf{13.54}_{\pm \mathbf{0.28}}$ | $\mathbf{16.03}_{\pm \mathbf{0.46}}$ | $\mathbf{19.33}_{\pm \mathbf{1.14}}$ | $\mathbf{21.59}_{\pm \mathbf{0.32}}$ |
| | (+7.16%) | (+4.12%) | (-2.89%) | (-21.66%) | **(-53.46%)** | **(-60.18%)** | **(-69.44%)** | **(-72.29%)** |

Table 4: Test error rates of ResNet-18 on CIFAR-10. Subscripts denote standard deviations, and bracketed numbers denote relative gains with respect to MP. Unpruned models have 8.68% error rate.

| | 10.30% | 6.33% | 3.89% | 2.40% | 1.48% | 0.92% | 0.57% | 0.36% |
|---|---|---|---|---|---|---|---|---|
| MP (baseline) | $8.18_{\pm 0.33}$ | $\mathbf{8.74}_{\pm \mathbf{0.15}}$ | $9.82_{\pm 0.18}$ | $\mathbf{11.28}_{\pm \mathbf{0.30}}$ | $14.31_{\pm 0.18}$ | $18.56_{\pm 0.36}$ | $22.93_{\pm 0.93}$ | $26.77_{\pm 1.04}$ |
| LAP | $\mathbf{8.09}_{\pm \mathbf{0.10}}$ | $8.97_{\pm 0.22}$ | $\mathbf{9.74}_{\pm \mathbf{0.15}}$ | $11.35_{\pm 0.20}$ | $\mathbf{13.73}_{\pm \mathbf{0.24}}$ | $\mathbf{16.29}_{\pm \mathbf{0.29}}$ | $\mathbf{20.22}_{\pm \mathbf{0.53}}$ | $\mathbf{22.45}_{\pm \mathbf{0.64}}$ |
| | **(-1.08%)** | (+2.59%) | **(-0.81%)** | (+0.64%) | **(-4.08%)** | **(-12.23%)** | **(-11.82%)** | **(-15.82%)** |
| LAP-forward | $8.19_{\pm 0.15}$ | $9.17_{\pm 0.07}$ | $10.32_{\pm 0.27}$ | $12.38_{\pm 0.30}$ | $15.31_{\pm 0.62}$ | $18.56_{\pm 0.88}$ | $21.09_{\pm 0.53}$ | $23.89_{\pm 0.46}$ |
| | (+0.12%) | (+4.85%) | (+5.09%) | (+9.79%) | (+6.96%) | (-0.02%) | (-8.04%) | (-10.44%) |

Table 5: Top-5 test error rates of VGG-19 on Tiny-ImageNet. Subscripts denote standard deviations, and bracketed numbers denote relative gains with respect to MP. Unpruned models have 36.89% error rate. Top-1 test error rates are presented in Table 10.

| | 12.16% | 10.34% | 8.80% | 7.48% | 6.36% | 5.41% | 4.61% | 3.92% |
|---|---|---|---|---|---|---|---|---|
| MP (baseline) | $36.40_{\pm 1.31}$ | $37.37_{\pm 1.08}$ | $38.40_{\pm 1.30}$ | $40.23_{\pm 1.26}$ | $42.68_{\pm 1.97}$ | $45.83_{\pm 2.76}$ | $49.79_{\pm 2.67}$ | $56.15_{\pm 5.14}$ |
| LAP | $\mathbf{36.01}_{\pm \mathbf{1.31}}$ | $\mathbf{37.03}_{\pm \mathbf{0.90}}$ | $\mathbf{38.20}_{\pm \mathbf{1.61}}$ | $\mathbf{39.36}_{\pm \mathbf{1.30}}$ | $40.95_{\pm 1.46}$ | $\mathbf{43.14}_{\pm \mathbf{1.33}}$ | $\mathbf{45.29}_{\pm \mathbf{1.80}}$ | $\mathbf{48.34}_{\pm \mathbf{0.30}}$ |
| | **(-1.07%)** | **(-0.90%)** | **(-0.52%)** | **(-2.16%)** | (-4.05%) | **(-5.87%)** | **(-9.02%)** | **(-13.92%)** |
| LAP-forward | $36.98_{\pm 1.04}$ | $37.35_{\pm 0.90}$ | $38.49_{\pm 1.10}$ | $39.57_{\pm 0.97}$ | $\mathbf{40.94}_{\pm \mathbf{1.49}}$ | $43.30_{\pm 1.57}$ | $45.76_{\pm 1.37}$ | $48.95_{\pm 1.70}$ |
| | (+1.58%) | (-0.04%) | (+0.24%) | (-1.65%) | **(-4.06%)** | (-5.53%) | (-8.08%) | (-12.84%) |

Table 6: Top-5 test error rates of ResNet-50 on Tiny-ImageNet. Subscripts denote standard deviations, and bracketed numbers denote relative gains with respect to MP. Unpruned models have 23.19% error rate. Top-1 test error rates are presented in Table 11.

| | 6.52% | 4.74% | 3.45% | 2.51% | 1.83% | 1.34% | 0.98% | 0.72% |
|---|---|---|---|---|---|---|---|---|
| MP (baseline) | $23.88_{\pm 0.27}$ | $24.99_{\pm 0.34}$ | $26.84_{\pm 0.39}$ | $29.54_{\pm 0.58}$ | $34.04_{\pm 0.48}$ | $40.19_{\pm 0.36}$ | $45.13_{\pm 0.57}$ | $59.18_{\pm 16.31}$ |
| LAP | $\mathbf{23.64}_{\pm \mathbf{0.40}}$ | $\mathbf{24.91}_{\pm \mathbf{0.25}}$ | $\mathbf{26.52}_{\pm \mathbf{0.38}}$ | $\mathbf{28.84}_{\pm \mathbf{0.43}}$ | $\mathbf{33.71}_{\pm \mathbf{0.58}}$ | $\mathbf{39.07}_{\pm \mathbf{0.45}}$ | $\mathbf{43.05}_{\pm \mathbf{0.97}}$ | $46.16_{\pm 1.04}$ |
| | **(-1.00%)** | **(-0.34%)** | **(-1.17%)** | **(-2.38%)** | **(-0.98%)** | **(-2.79%)** | **(-4.61%)** | (-22.00%) |
| LAP-forward | $24.26_{\pm 0.48}$ | $24.92_{\pm 0.41}$ | $27.66_{\pm 0.55}$ | $30.93_{\pm 0.81}$ | $35.90_{\pm 1.24}$ | $39.99_{\pm 0.58}$ | $43.42_{\pm 0.52}$ | $\mathbf{45.45}_{\pm \mathbf{0.78}}$ |
| | (+1.57%) | (-0.30%) | (+3.08%) | (+4.71%) | (+5.46%) | (-0.48%) | (-3.79%) | **(-23.19%)** |

Table 7: Top-5 test error rates of WRN-16-8 on Tiny-ImageNet. Subscripts denote standard deviations, and bracketed numbers denote relative gains with respect to MP. Unpruned models have 25.77% error rate. Top-1 test error rates are presented in Table 12.

| | 12.22% | 8.85% | 6.41% | 4.65% | 3.37% | 2.45% | 1.77% | 1.29% |
|---|---|---|---|---|---|---|---|---|
| MP (baseline) | $25.27_{\pm 0.73}$ | $26.79_{\pm 0.87}$ | $28.84_{\pm 1.04}$ | $\mathbf{31.91}_{\pm \mathbf{0.80}}$ | $37.01_{\pm 1.42}$ | $42.89_{\pm 2.43}$ | $51.10_{\pm 2.59}$ | $59.73_{\pm 2.85}$ |
| LAP | $\mathbf{24.99}_{\pm \mathbf{0.85}}$ | $\mathbf{26.55}_{\pm \mathbf{1.45}}$ | $\mathbf{28.68}_{\pm \mathbf{1.17}}$ | $32.22_{\pm 2.51}$ | $\mathbf{35.82}_{\pm \mathbf{2.06}}$ | $\mathbf{41.37}_{\pm \mathbf{3.07}}$ | $45.43_{\pm 4.48}$ | $\mathbf{51.83}_{\pm \mathbf{1.91}}$ |
| | **(-1.12%)** | **(-0.87%)** | **(-0.58%)** | (+0.98%) | **(-3.22%)** | **(-3.55%)** | (-11.10%) | **(-13.22%)** |
| LAP-forward | $26.30_{\pm 0.88}$ | $28.52_{\pm 2.13}$ | $30.98_{\pm 1.39}$ | $34.72_{\pm 1.82}$ | $38.41_{\pm 2.48}$ | $42.02_{\pm 2.46}$ | $\mathbf{45.10}_{\pm \mathbf{1.80}}$ | $51.92_{\pm 1.94}$ |
| | (+4.08%) | (+6.48%) | (+7.42%) | (+8.83%) | (+3.79%) | (-2.02%) | **(-11.74%)** | (-13.07%) |

## 4 CONCLUSION

In this work, we interpret magnitude-based pruning as a solution to the minimization of the Frobenius distortion of a single layer operation incurred by pruning. Based on this framework, we consider the minimization of the Frobenius distortion of multi-layer operation, and propose a novel lookahead pruning (LAP) scheme as a computationally efficient algorithm to solve the optimization. Although LAP was motivated from linear networks, it extends to nonlinear networks which indeed minimizes the root mean square lookahead distortion assuming i.i.d. activations. We empirically show its effectiveness on networks with nonlinear activation functions, and test the algorithm on various network architectures including VGG, ResNet and WRN, where LAP consistently performs better than MP.

**Acknowledgments.** We thank Seunghyun Lee for providing helpful feedbacks and suggestions in preparing the early version of the manuscript. JL also gratefully acknowledges Jungseul Ok and Phillip M. Long for enlightening discussions about theoretical natures of neural network pruning. This research was supported by the Engineering Research Center Program through the National Research Foundation of Korea (NRF), funded by the Korean Government MSIT (NRF-2018R1A5A1059921).

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

# A    EXPERIMENTAL SETUP

**Models and datasets.**    We consider four neural network architectures: (1) The fully-connected network (FCN) under consideration is composed of four hidden layers, each with 500 hidden neurons. (2) The convolutional network (Conv-6) consists of six convolutional layers, followed by a fully-connected classifier with two hidden layers with 256 hidden neurons each; this model is identical to that appearing in the work of Frankle & Carbin (2019) suggested as a scaled-down variant of VGG.[5] (3) VGGs of depths $\{11, 16, 19\}$ were used, with an addition of batch normalization layers after each convolutional layers, and a reduced number of fully-connected layers from three to one.[6] (4) ResNets with depth $\{18, 50\}$ are used. (5) Wide ResNets with depth 16 and widening factor 8 is used. All networks are initialized via the method of Glorot & Bengio (2010), except for ResNets and WRN. We use the ReLU activation function except for the experiments in Section 3.1. We focus on image classification tasks. FCN is trained with MNIST dataset (Lecun et al., 1998), Conv-6, VGG-$\{11, 16, 19\}$ and ResNet-18 are trained with CIFAR-10 dataset (Krizhevsky & Hinton, 2009), and VGG-19, ResNet-50, WRN-16-8 ware trained with Tiny-ImageNet dataset.

**Optimizers and hyperparameters.**    We use Adam optimizer (Kingma & Ba, 2015) with batch size 60. We use a learning rate of $1.2 \cdot 10^{-3}$ for FCN and $3 \cdot 10^{-4}$ for all other models. For FCN, we use [50k, 50k] for the initial training phase and retraining phase. For Conv-6, we use [30k, 20k] steps. For VGG-11 and ResNet-18, we use [35k, 25k] steps. For VGG-16, we use [50k, 35k]. For VGG-19, ResNet-50, and WRN-16-8 we use [60k, 40k]. We do not use any weight decay, learning rate scheduling, or regularization.

**Sparsity levels.**    To determine the layerwise pruning ratio, we largely follow the the guidelines of Han et al. (2015); Frankle & Carbin (2019): For integer values of $\tau$, we keep $p^\tau$ fraction of weights in all convolutional layers and $q^\tau$ fraction in all fully-connected layers, except for the last layer where we use $(1 + q)/2$ instead. For FCN, we use $(p, q) = (0, 0.5)$. For Conv-6, VGGs ResNets, and WRN, we use $(0.85, 0.8)$. For ResNet-$\{18, 50\}$, we do not prune the first convolutional layer. The range of sparsity for reported figures in all tables is decided as follows: we start from $\tau$ where test error rate starts falling below that of an unpruned model and report the results at $\tau, \tau + 1, \tau + 2, \ldots$ for FCN and Conv-6, $\tau, \tau + 2, \tau + 4, \ldots$ for VGGs, ResNet-50, and WRN, and $\tau, \tau + 3, \tau + 6, \ldots$ for ResNet-18.

# B    ADDITIONAL VGG EXPERIMENTS

Table 8: Test error rates of VGG-11 on CIFAR-10. Subscripts denote standard deviations, and bracketed numbers denote relative gains with respect to MP. Unpruned models have 11.51% error rate.

| | 16.74% | 12.10% | 8.74% | 6.32% | 4.56% | 3.30% | 2.38% | 1.72% |
|---|---|---|---|---|---|---|---|---|
| MP (baseline) | $11.41_{\pm 0.24}$ | $12.38_{\pm 0.14}$ | $13.54_{\pm 0.35}$ | $16.08_{\pm 1.13}$ | $19.76_{\pm 1.67}$ | $28.12_{\pm 3.45}$ | $45.38_{\pm 11.69}$ | $55.97_{\pm 15.99}$ |
| LAP | $\mathbf{11.19}_{\pm \mathbf{0.15}}$ | $\mathbf{11.79}_{\pm \mathbf{0.44}}$ | $\mathbf{12.95}_{\pm \mathbf{0.14}}$ | $\mathbf{13.95}_{\pm \mathbf{0.17}}$ | $15.59_{\pm 0.35}$ | $20.96_{\pm 6.02}$ | $22.00_{\pm 1.09}$ | $28.96_{\pm 3.30}$ |
| | **(-1.96%)** | **(-4.78%)** | **(-4.39%)** | **(-13.25%)** | (-21.13%) | (-25.47%) | (-51.52%) | (-48.25%) |
| LAP-forward | $11.47_{\pm 0.30}$ | $12.33_{\pm 0.12}$ | $13.15_{\pm 0.22}$ | $13.96_{\pm 0.25}$ | $\mathbf{15.42}_{\pm \mathbf{0.21}}$ | $\mathbf{18.22}_{\pm \mathbf{0.69}}$ | $\mathbf{21.74}_{\pm \mathbf{1.59}}$ | $\mathbf{25.85}_{\pm \mathbf{1.40}}$ |
| | (+0.56%) | (-0.44%) | (-2.87%) | (-13.18%) | **(-21.97%)** | **(-35.20%)** | **(-52.10%)** | **(-53.82%)** |

Table 9: Test error rates of VGG-16 on CIFAR-10. Subscripts denote standard deviations, and bracketed numbers denote relative gains with respect to MP. Unpruned models have 9.33% error rate.

| | 10.28% | 7.43% | 5.37% | 3.88% | 2.80% | 2.03% | 1.46% | 1.06% |
|---|---|---|---|---|---|---|---|---|
| MP (baseline) | $9.55_{\pm 0.11}$ | $10.78_{\pm 0.45}$ | $13.42_{\pm 2.19}$ | $17.83_{\pm 3.08}$ | $26.61_{\pm 4.91}$ | $48.87_{\pm 5.85}$ | $69.39_{\pm 11.85}$ | $83.47_{\pm 5.60}$ |
| LAP | $\mathbf{9.35}_{\pm \mathbf{0.18}}$ | $\mathbf{10.07}_{\pm \mathbf{0.19}}$ | $\mathbf{11.52}_{\pm \mathbf{0.26}}$ | $\mathbf{12.57}_{\pm \mathbf{0.34}}$ | $\mathbf{14.23}_{\pm \mathbf{0.27}}$ | $\mathbf{17.01}_{\pm \mathbf{1.46}}$ | $25.03_{\pm 2.08}$ | $32.45_{\pm 12.20}$ |
| | **(-2.05%)** | **(-6.59%)** | **(-14.21%)** | **(-29.50%)** | **(-46.52%)** | **(-65.19%)** | (-63.92%) | (-61.12%) |
| LAP-forward | $9.45_{\pm 0.17}$ | $10.40_{\pm 0.20}$ | $11.33_{\pm 0.15}$ | $13.09_{\pm 0.21}$ | $14.61_{\pm 0.25}$ | $17.10_{\pm 0.19}$ | $\mathbf{22.39}_{\pm \mathbf{0.74}}$ | $\mathbf{24.99}_{\pm \mathbf{0.49}}$ |
| | (-1.03%) | (-3.49%) | (-15.60%) | (-26.56%) | (-45.08%) | (-65.02%) | **(-67.74%)** | **(-70.06%)** |

---

[5]Convolutional layers are organized as $[64, 64] - \mathsf{MaxPool} - [128, 128] - \mathsf{MaxPool} - [256, 256]$.

[6]This is a popular configuration of VGG for CIFAR-10 (Liu et al., 2019; Frankle & Carbin, 2019)

## C Top-1 error rates for Tiny-ImageNet experiments

Table 10: Top-1 test error rates of VGG-19 on Tiny-ImageNet. Subscripts denote standard deviations, and bracketed numbers denote relative gains with respect to MP. Unpruned models have 64.55% error rate.

| | 12.16% | 10.34% | 8.80% | 7.48% | 6.36% | 5.41% | 4.61% | 3.92% |
|---|---|---|---|---|---|---|---|---|
| MP (baseline) | $63.35_{\pm1.44}$ | $64.43_{\pm1.05}$ | $\mathbf{65.44_{\pm1.31}}$ | $67.09_{\pm1.04}$ | $69.40_{\pm1.40}$ | $72.36_{\pm2.09}$ | $75.35_{\pm1.75}$ | $79.98_{\pm3.28}$ |
| LAP | $\mathbf{63.15_{\pm1.52}}$ (-0.31%) | $\mathbf{63.91_{\pm1.38}}$ (-0.80%) | $65.56_{\pm1.42}$ (+0.18%) | $\mathbf{66.56_{\pm0.93}}$ (-0.80%) | $\mathbf{68.40_{\pm1.08}}$ (-1.44%) | $\mathbf{70.45_{\pm0.67}}$ (-2.63%) | $\mathbf{72.16_{\pm1.62}}$ (-4.24%) | $\mathbf{75.05_{\pm0.29}}$ (-6.17%) |
| LAP-forward | $64.22_{\pm1.11}$ (+1.38%) | $64.77_{\pm0.96}$ (+0.53%) | $65.63_{\pm1.21}$ (+0.28%) | $67.03_{\pm1.23}$ (-0.09%) | $68.52_{\pm1.39}$ (-1.26%) | $70.55_{\pm1.21}$ (-2.50%) | $73.13_{\pm0.97}$ (-2.95%) | $75.71_{\pm1.33}$ (-5.34%) |

Table 11: Top-1 test error rates of ResNet-50 on Tiny-ImageNet. Subscripts denote standard deviations, and bracketed numbers denote relative gains with respect to MP. Unpruned models have 47.50% error rate.

| | 6.52% | 4.74% | 3.45% | 2.51% | 1.83% | 1.34% | 0.98% | 0.72% |
|---|---|---|---|---|---|---|---|---|
| MP (baseline) | $\mathbf{48.18_{\pm0.39}}$ | $\mathbf{49.85_{\pm0.30}}$ | $52.28_{\pm0.24}$ | $55.46_{\pm0.57}$ | $60.51_{\pm0.39}$ | $66.60_{\pm0.42}$ | $70.75_{\pm0.33}$ | $80.02_{\pm8.94}$ |
| LAP | $48.27_{\pm0.13}$ (+0.20%) | $49.96_{\pm0.26}$ (+0.22%) | $\mathbf{51.92_{\pm0.21}}$ (-0.69%) | $\mathbf{54.91_{\pm0.45}}$ (-0.99%) | $\mathbf{60.31_{\pm0.18}}$ (-0.34%) | $\mathbf{65.46_{\pm0.27}}$ (-1.71%) | $\mathbf{69.13_{\pm0.91}}$ (-2.29%) | $71.81_{\pm0.84}$ (-10.26%) |
| LAP-forward | $48.69_{\pm0.52}$ (+1.05%) | $50.25_{\pm0.26}$ (+0.79%) | $53.55_{\pm0.42}$ (+2.42%) | $57.59_{\pm0.61}$ (+3.84%) | $62.74_{\pm0.87}$ (+3.69%) | $66.59_{\pm0.89}$ (-0.02%) | $69.55_{\pm0.25}$ (-1.69%) | $\mathbf{71.49_{\pm0.57}}$ (-10.67%) |

Table 12: Top-1 test error rates of WRN-16-8 on Tiny-ImageNet. Subscripts denote standard deviations, and bracketed numbers denote relative gains with respect to MP. Unpruned models have 51.85% error rate.

| | 12.22% | 8.85% | 6.41% | 4.65% | 3.37% | 2.45% | 1.77% | 1.29% |
|---|---|---|---|---|---|---|---|---|
| MP (baseline) | $50.38_{\pm1.00}$ | $52.64_{\pm0.84}$ | $55.23_{\pm1.13}$ | $\mathbf{58.79_{\pm0.81}}$ | $64.11_{\pm1.23}$ | $69.22_{\pm2.03}$ | $75.90_{\pm2.03}$ | $81.83_{\pm2.17}$ |
| LAP | $\mathbf{49.85_{\pm1.19}}$ (-1.04%) | $\mathbf{52.33_{\pm1.69}}$ (-0.60%) | $\mathbf{54.96_{\pm1.26}}$ (-0.49%) | $59.06_{\pm2.40}$ (+0.46%) | $\mathbf{62.68_{\pm1.57}}$ (-2.23%) | $\mathbf{67.82_{\pm2.39}}$ (-2.02%) | $\mathbf{71.30_{\pm3.65}}$ (-6.06%) | $\mathbf{76.51_{\pm1.54}}$ (-6.50%) |
| LAP-forward | $51.86_{\pm1.14}$ (+2.95%) | $54.77_{\pm2.37}$ (+4.05%) | $57.65_{\pm1.75}$ (+4.38%) | $61.84_{\pm1.39}$ (+5.18%) | $65.30_{\pm2.16}$ (+1.85%) | $69.03_{\pm2.46}$ (-0.27%) | $71.75_{\pm1.66}$ (-5.46%) | $77.00_{\pm1.23}$ (-5.91%) |

## D    NP-HARDNESS OF EQ. (3)

In this section, we show that the optimization in Eq. (3) is NP-hard by showing the reduction from the following binary quadratic programming which is NP-hard (Murty & Kabadi, 1987):

$$\min_{x \in \{0,1\}^n} x^T A x \tag{9}$$

for some symmetric matrix $A \in \mathbb{R}^{n \times n}$. Without loss of generality, we assume that the minimum eigenvalue of $A$ (denoted with $\lambda$) is negative; if not, Eq. (9) admits a trivial solution $x = (0, \dots, 0)$.

Assuming $\lambda < 0$, Eq. (9) can be reformulated as:

$$\min_{x \in \{0,1\}^n} x^T H x + \lambda \sum_i x_i \tag{10}$$

where $H = A - \lambda I$. Here, one can easily observe that the above optimization can be solved by solving the below optimization for $s = 1, \dots, n$

$$\min_{x \in \{0,1\}^n : \sum_i x_i = s} x^T H x \tag{11}$$

Finally, we introduce the below equality

$$x^\top H x = x^\top U \Lambda U^\top x \tag{12}$$

$$= \|\sqrt{\Lambda} U^\top x\|_F^2 \tag{13}$$

$$= \|\sqrt{\Lambda} U^\top x\|_F^2 \tag{14}$$

$$= \|\sqrt{\Lambda} U^\top \mathbf{1} - \sqrt{\Lambda} U^\top \left((\mathbf{1} - x) \odot \mathbf{1}\right)\|_F^2 \tag{15}$$

where $\mathbf{1}$ denotes a vector of ones, $U$ is a matrix consisting of the eigenvectors of $H$ as its column vectors, and $\Lambda$ is a diagonal matrix with corresponding (positive) eigenvalues of $H$ as its diagonal elements. The above equality shows that Eq. (11) is a special case of Eq. (3) by choosing $W_1 = \sqrt{\Lambda} U^\top, W_2 = \mathbf{1}, W_3 = 1$ and $M = \mathbf{1} - x$. This completes the reduction from Eq. (9) to Eq. (3).

## E    DERIVATION OF EQ. (5)

In this section, we provide a derivation of Eq. (5) for the fully-connected layers. The convolutional layers can be handled similarly by substituting the multiplications in Eqs. (16) and (17) by the convolutions.

The Jacobian matrix of the linear operator correponding to a fully-connected layer is the weight matrix itself, i.e. $\mathcal{J}(W_i) = W_i$. From this, lookahead distortion can be reformulated as

$$\mathcal{L}_i(w) = \left\|W_{i+1} W_i W_{i-1} - W_{i+1} W_i|_{w=0} W_{i-1}\right\|_F. \tag{16}$$

Now, we decompose the matrix product $W_{i+1} W_i W_{i-1}$ in terms of entries of $W_i$ as below:

$$W_{i+1} W_i W_{i-1} = \sum_{j,k} W_i[k,j] W_{i+1}[:,k] W_{i-1}[j,:] \tag{17}$$

where $W_i[k,j], W_{i+1}[:,k]$, and $W_{i-1}[j,:]$ denote $(j,k)$-th element of $W_i$, $k$-th column of $W_{i+1}$, and $j$-th row of $W_{i-1}$, respectively. The contribution of a single entry $w := W_i[k,j]$ to the product $W_{i+1} W_i W_{i-1}$ is equivalent to $w \cdot W_{i+1}[:,k] W_{i-1}[j,:]$. Therefore, in terms of the Frobenius distortion, we conclude that

$$\mathcal{L}_i(w) = \left\|w \cdot W_{i+1}[:,k] W_{i-1}[j,:]\right\|_F = |w| \cdot \left\|W_{i-1}[j,:]\right\|_F \cdot \left\|W_{i+1}[:,k]\right\|_F,$$

which completes the derivation of Eq. (5) for fully-connected layers.

## F  LAP-ACT: IMPROVING LAP USING TRAINING DATA

Recall two observations made from the example of two-layer fully connected network with ReLU activation appearing in Section 2.1: LAP is designed to reflect the lack of knowledge about the training data at the pruning phase; once the activation probability of each neuron can be estimated, it is possible to refine LAP to account for this information.

In this section, we continue our discussion on the second observation. In particular, we study an extension of LAP called *lookahead pruning with activation* (LAP-act) which prunes the weight with smallest value of

$$\widehat{\mathcal{L}}_i(w) := |\widehat{w}| \cdot \left\| \widehat{W}_{i-1}[j, :] \right\|_F \cdot \left\| \widehat{W}_{i+1}[:, k] \right\|_F. \tag{18}$$

Here, $\widehat{W}_i$ is a scaled version of $W_i$ and $\widehat{w}$ is the corresponding scaled value of $w$, defined by

$$\widehat{W}_i[j, :] := \left( \sum_{k \in I_{i,j}} \sqrt{p_k} \right) \cdot W_i[j, :], \tag{19}$$

where $I_{i,j}$ denotes the set of ReLU indices in the $j$-th output neuron/channel of $i$-th layer. For example, $I_{i,j} = \{j\}$ for fully connected layers and $I_{i,j}$ is a set of ReLU indices in the $j$-th channel for convolutional layers. Also, $p_k$ denotes the $k$-th ReLU's probability of activation, which can be estimated by passing the training data.

We derive LAP-act (Eq. (18)) in Appendix F.1 and perform preliminary empirical validations in Appendix F.2 with using optimal brain damage (OBD) as a baseline. We also evaluate a variant of LAP using Hessian scores of OBD instead of magnitude scores. It turns out that in the small networks (FCN, Conv-6), LAP-act outperforms OBD.

### F.1  DERIVATION OF LAP-ACT

Consider a case where one aims to prune a connection of a network with ReLU, i.e.,

$$x \mapsto \mathcal{J}(W_L)\sigma(\mathcal{J}(W_{L-1}) \cdots \sigma(\mathcal{J}(W_1)x) \cdots), \tag{20}$$

where $\sigma(x) = \max\{0, x\}$ is applied entrywise. Under the over-parametrized scenario, zeroing out a single weight may alter the activation pattern of connected neurons with only negligible probability, which allows one to decouple the probability of activation of each neuron from the act of pruning each connection. From this observation, we first construct the below random distortion, following the philosophy of the linear lookahead distortion Eq. (4)

$$\widetilde{\mathcal{L}}_i(w) := \|\widetilde{\mathcal{J}}(W_{i+1})(\widetilde{\mathcal{J}}(W_i) - \widetilde{\mathcal{J}}(W_i|_{w=0}))\widetilde{\mathcal{J}}(W_{i-1})\|_F \tag{21}$$

where $\widetilde{\mathcal{J}}(W_i)$ denotes a random matrix where $\widetilde{\mathcal{J}}(W_i)[k, :] = g_i[k] \cdot \mathcal{J}(W_i)[k, :]$ and $g_i[k]$ is a 0-1 random variable corresponding to the activation, i.e., $g_i[k] = 1$ if and only if the $k$-th output, i.e., ReLU, of the $i$-th layer is activated. However, directly computing the expected distortion with respect to the real activation distribution might be computationally expensive. To resolve this issue, we approximate the root mean square lookahead distortion by applying the mean-field approximation to the activation probability of neurons, i.e., all activations are assumed to be independent, as

$$\sqrt{\mathbb{E}_{g \sim p(g)}[\widetilde{\mathcal{L}}_i(w)^2]} \approx \sqrt{\mathbb{E}_{g \sim \prod_{i,k} p(g_i[k])}[\widetilde{\mathcal{L}}_i(w)^2]} =: \widehat{\mathcal{L}}_i(w) \tag{22}$$

where $g = [g_i]_i$, $p(g)$ denotes the empirical activation distribution of all neurons and $\prod_{i,k} p(g_i[k])$ denotes the mean-field approximation of $p(g)$. Indeed, the lookahead distortion with ReLU non-linearity (Eq. (22)) or three-layer blocks consisting only of the fully-connected layers and the convolutional layers can be easily computed by using the rescaled weight matrix $\widehat{W}_i$:

$$\widehat{W}_i[j, :] := \left( \sum_{k \in I_{i,j}} \sqrt{p(g_i[k] = 1)} \right) \cdot W_i[j, :] \tag{23}$$

where $I_{i,j}$ denotes the set of ReLU indices in the $j$-th output neuron/channel of $i$-th layer. For example, $I_{i,j} = \{j\}$ for fully connected layers and $I_{i,j}$ is a set of ReLU indices in the $j$-th channel

for convolutional layers. Finally, for an edge $w$ connected to the $j$-th input neuron/channel and the $k$-th output neuron/channel of the $i$-th layer, Eq. (22) reduces to

$$\widehat{\mathcal{L}}_i(w) = |\widehat{w}| \cdot \left\|\widehat{W}_{i-1}[j,:]\right\|_F \cdot \left\|\widehat{W}_{i+1}[:,k]\right\|_F \tag{24}$$

where $\widehat{w}$ denotes the rescaled value of $w$. This completes the derivation of Eq. (18).

### F.2 EXPERIMENTS WITH LAP-ACT

We compare the performance of three algorithms utilizing training data at the pruning phase: optimal brain damage (OBD) which approximates the loss via second order Taylor seris approximation with the Hessian diagonal (LeCun et al., 1989), LAP using OBD instead of weight magnitudes (OBD+LAP), and LAP-act as described in this section. We compare the performances of three algorithms under the same experimental setup as in Section 3.2. To compute the Hessian diagonal for OBD and OBD+LAP, we use a recently introduced software package called "BackPACK," (Dangel et al., 2020), which is the only open-source package supporting an efficient of Hessians, up to our knowledge. Note that the algorithms evaluated in this section are also evaluated for global pruning experiments in Appendix I.

The experimental results for FCN and Conv-6 are presented in Tables 13 and 14. Comparing to algorithms relying solely on the model parameters for pruning (MP/LAP in Tables 1 and 2), we observe that OBD performs better in general, especially in the high sparsity regime. This observation is coherent to the findings of LeCun et al. (1989). Intriguingly, however, we observe that applying lookahead critertion to OBD (OBD+LAP) significantly enhances to OBD significantly enhances the performance in the high sparsity regime. We hypothesize that LAP helps capturing a correlation among scores (magnitude or Hessian-based) of adjacent layers. Also, we observe that LAP-act consistently exhibits a better performance compared to OBD. This result is somewhat surprising, in the sense that LAP-act only utilizes (easier-to-estimate) information about activation probabilities of each neuron to correct lookahead distortion.

The average running time of OBD, OBD+LAP, and LAP-act is summarized in Table 15. We use Xeon E5-2630v4 2.20GHz for pruning edges, and additionally used a single NVidia GeForce GTX-1080 for the computation of Hessian diagonals (used for OBD, OBD+LAP) and activation probabiility (for LAP-act). We observe that LAP-act runs in a significantly less running time than OBD/OBD+LAP, and the gap widens as the number of parameters and the dimensionality of the dataset increases (from MNIST to CIFAR-10).

Table 13: Test error rates of FCN on MNIST. Subscripts denote standard deviations, and bracketed numbers denote relative gains with respect to OBD. Unpruned models achieve $1.98\%$ error rate.

|  | 6.36% | 3.21% | 1.63% | 0.84% | 0.43% | 0.23% | 0.12% |
|---|---|---|---|---|---|---|---|
| OBD (baseline) | $1.87_{\pm 0.05}$ | $2.07_{\pm 0.13}$ | $2.51_{\pm 0.10}$ | $3.07_{\pm 0.12}$ | $4.08_{\pm 0.14}$ | $5.66_{\pm 0.39}$ | $11.01_{\pm 1.71}$ |
| OBD+LAP | $1.81_{\pm 0.05}$ | $2.18_{\pm 0.13}$ | $2.52_{\pm 0.14}$ | $3.48_{\pm 0.14}$ | $4.16_{\pm 0.35}$ | $5.88_{\pm 0.51}$ | $8.65_{\pm 0.56}$ |
|  | (-3.42%) | (+5.31%) | (+0.48%) | (+13.35%) | (+1.91%) | (+3.81%) | (-21.41%) |
| LAP-act | $\mathbf{1.78_{\pm 0.07}}$ | $\mathbf{1.85_{\pm 0.09}}$ | $\mathbf{2.21_{\pm 0.13}}$ | $\mathbf{2.73_{\pm 0.04}}$ | $\mathbf{3.50_{\pm 0.35}}$ | $\mathbf{4.74_{\pm 0.21}}$ | $\mathbf{7.99_{\pm 0.19}}$ |
|  | (-4.60%) | (-10.63%) | (-12.11%) | (-11.13%) | (-14.31%) | (-16.21%) | (-27.48%) |

Table 14: Test error rates of Conv-6 on CIFAR-10. Subscripts denote standard deviations, and bracketed numbers denote relative gains with respect to OBD. Unpruned models achieve $11.97\%$ error rate.

|  | 10.62% | 8.86% | 7.39% | 6.18% | 5.17% | 4.32% | 3.62% |
|---|---|---|---|---|---|---|---|
| OBD (baseline) | $\mathbf{12.10_{\pm 0.21}}$ | $12.81_{\pm 0.61}$ | $13.18_{\pm 0.26}$ | $14.28_{\pm 0.55}$ | $15.54_{\pm 0.40}$ | $16.83_{\pm 0.27}$ | $19.14_{\pm 0.32}$ |
| OBD+LAP | $12.51_{\pm 0.21}$ | $13.22_{\pm 0.48}$ | $13.68_{\pm 0.57}$ | $14.31_{\pm 0.36}$ | $15.09_{\pm 0.36}$ | $\mathbf{16.31_{\pm 0.51}}$ | $\mathbf{17.29_{\pm 0.47}}$ |
|  | (+3.41%) | (+3.20%) | (+2.23%) | (+0.18%) | (-2.90%) | (-3.13%) | (-9.65%) |
| LAP-act | $12.11_{\pm 0.12}$ | $\mathbf{12.72_{\pm 0.11}}$ | $\mathbf{12.92_{\pm 0.48}}$ | $\mathbf{13.45_{\pm 0.25}}$ | $\mathbf{14.86_{\pm 0.13}}$ | $16.47_{\pm 0.36}$ | $18.48_{\pm 0.33}$ |
|  | (+0.12%) | (-0.69%) | (-3.47%) | (-5.87%) | (-4.40%) | (-2.13%) | (-3.46%) |

Table 15: Computation time of OBD, OBD+LAP and LAP-act (averaged over 100 trials).

|  | FCN | Conv-6 |
|---|---|---|
| OBD (baseline) | 11.38 (s) | 167.87 (s) |
| OBD+LAP | 11.61 (s) | 168.03 (s) |
| LAP-act | 6.28 (s) | 8.95 (s) |
| # weight parameters | 1.15M | 2.26M |

## G  COMPUTATIONAL COST OF LOOKING AHEAD

In this section, we briefly describe how a computation of lookahead distortion Eq. (5) can be done efficiently, and provide experimental comparisons of average computation times for MP and LAP. It turns out that most of the computational load for LAP comes from the sorting procedure, and tensor operations introduce only a minimal overhead.

MP comprises of three steps: (1) computing the absolute value of the tensor, (2) sorting the absolute values, and (3) selecting the cut-off threshold and zero-ing out the weights under the threshold. Steps (2) and (3) remain the same in LAP, and typically takes $\mathcal{O}(n \log n)$ steps ($n$ denotes the number of parameters in a layer). On the other hand, Step (1) is replaced by computing the lookahead distortion

$$\mathcal{L}_i(w) = |w| \cdot \|W_{i-1}[j, :]\|_F \, \|W_{i+1}[:, k]\|_F$$

for each parameter $w$. Fortunately, this need not be computed separately for each parameter. Indeed, one can perform tensor operations to compute the squared lookahead distortion, which has the same ordering with lookahead distortion. For fully-connected layers with 2-dimensional Jacobians, the squared lookahead distortion for $W_{i+1} \in \mathbb{R}^{d_{i+1} \times d_i}, W_i \in \mathbb{R}^{d_i \times d_{i-1}}, W_{i-1} \in \mathbb{R}^{d_{i-1} \times d_{i-2}}$ is

$$\mathcal{L}^2(W_i) = (\mathbf{1}_{i+1} W_{i+1}^{\odot 2})^\top \odot (W_i^{\odot 2}) \odot (W_{i-1}^{\odot 2} \mathbf{1}_i)^\top, \tag{25}$$

where $\mathbf{1}_i$ denotes all-one matrix of size $d_{i-2} \times d_i$; multiplying $\mathbf{1}_i$ denotes summing operation along an axis and duplicating summed results into the axis, and $^{\odot 2}$ denotes the element-wise square operation. The case of convolutional layers can be handled similarly.

We note that an implementation of Eq. (25) is very simple. Indeed, the following PyTorch code segment calculates a lookahead score matrix:

```
def lookahead_score(W,W_prev,W_next):
    W_prev_sq = (W_prev ** 2).sum(dim=1)
    W_prev_mat = W_prev_sq.view(1,-1).repeat(W.size(0),1)

    W_next_sq = (W_next ** 2).sum(dim=0)
    W_next_mat = W_next_sq.view(-1,1).repeat(1,W.size(1))

    return (W**2)*W_prev_mat*W_next_mat
```

Combined with modern tensor computation frameworks, computing Eq. (25) does not introduce heavy overhead. To show this, we compare the computation time of MP and LAP for six neural networks in Table 16, where we fixed the layerwise pruning rate to be uniformly 90%. The codes are implemented with PyTorch, and the computations have taken place on 40 CPUs of Intel Xeon E5-2630v4 @ 2.20GHz. All figures are averaged over 100 trials.

We make two observations from Table 16. First, the time required for LAP did not exceed 150% of the time required for MP, confirming our claim on the computational benefits of LAP. Second, most of the added computation comes from considering the factors from batch normalization, without which the added computation load is $\approx 5\%$.

Table 16: Computation time of MP and LAP on FCN, Conv-6, VGG-{11,16,19}, ResNet-18. All figures are averaged over 100 independent trials. Bracketed numbers denote relative increments. Number of weight parameters denote the number of parameters that are the target of pruning.

|  | FCN | Conv-6 | VGG-11 | VGG-16 | VGG-19 | ResNet-18 |
|---|---|---|---|---|---|---|
| MP (baseline) | 46.23 (ms) | 108.92 (ms) | 542.95 (ms) | 865.91 (ms) | 1188.29 (ms) | 641.59 (ms) |
| LAP (w/o batchnorm) | 47.73 (ms) (+3.14%) | 116.74 (ms) (+7.18%) | 560.60 (ms) (+3.25%) | 912.47 (ms) (+5.28%) | 1241.55 (ms) (+4.48%) | 671.61 (ms) (+4.68%) |
| LAP | - - | - - | 805.98 (ms) (+48.44%) | 1213.24 (ms) (+40.11%) | 1653.02 (ms) (+39.19%) | 943.86 (ms) (+47.11%) |
| # weight parameters | 1.15M | 2.26M | 9.23M | 14.72M | 20.03M | 10.99M |

## H  LOOKAHEAD FOR CHANNEL PRUNING

In the main text, LAP is compared to MP in the context of unstructured pruning, where we do not impose any structural constraints on the set of connections to be pruned together. On the other hand, the magnitude-based pruning methods are also being used popularly as a baseline for channel pruning (Ye et al., 2018), which falls under the category of structured pruning.

MP in channel pruning is typically done by removing channels with smallest aggregated weight magnitudes; this aggregation can be done by either taking $\ell_1$-norm or $\ell_2$-norm of magnitudes. Similarly, we can consider channel pruning scheme based on an $\ell_1$ or $\ell_2$ aggregation of LAP distortions, which we will call LAP-$\ell_1$ and LAP-$\ell_2$ (as opposed to MP-$\ell_1$ and MP-$\ell_2$).

We compare the performances of LAP-based channel pruning methods to MP-based channel pruning methods, along with another baseline of random channel pruning (denoted with RP). We test with Conv-6 (Table 17) and VGG-19 (Table 18) networks on CIFAR-10 dataset. All reported figures are averaged over five trials, experimental settings are identical to the unstructure pruning experiments unless noted otherwise.

Similar to the case of unstructured pruning, we observe that LAP-based methods consistently outperform MP-based methods. Comparing $\ell_1$ with $\ell_2$ aggregation, we note that LAP-$\ell_2$ performs better than LAP-$\ell_1$ in both experiments, by a small margin. Among MP-based methods, we do not observe any similar dominance.

Table 17: Test error rates of Conv-6 on CIFAR-10 for channel pruning. Subscripts denote standard deviations, and bracketed numbers denote relative gains with respect to the best of MP-$\ell_1$ and MP-$\ell_2$. Unpruned models achieve 11.97% error rate.

|  | 34.40% | 24.01% | 16.81% | 11.77% | 8.24% | 5.76% | 4.04% | 2.82% |
|---|---|---|---|---|---|---|---|---|
| MP-$\ell_1$ | $12.11_{\pm0.38}$ | $12.55_{\pm0.44}$ | $13.62_{\pm0.44}$ | $16.85_{\pm1.14}$ | $20.05_{\pm0.61}$ | $23.98_{\pm0.92}$ | $27.75_{\pm0.89}$ | $37.56_{\pm2.16}$ |
| MP-$\ell_2$ | $11.97_{\pm0.39}$ | $12.66_{\pm0.24}$ | $14.17_{\pm0.53}$ | $16.69_{\pm1.08}$ | $20.09_{\pm0.96}$ | $24.61_{\pm1.94}$ | $28.30_{\pm1.47}$ | $35.18_{\pm1.80}$ |
| RP | $12.94_{\pm0.41}$ | $14.82_{\pm0.27}$ | $17.57_{\pm0.65}$ | $20.19_{\pm0.54}$ | $22.50_{\pm0.69}$ | $25.86_{\pm0.72}$ | $30.64_{\pm0.87}$ | $38.26_{\pm2.78}$ |
| LAP-$\ell_1$ | $12.08_{\pm0.28}$ (+0.87%) | $12.57_{\pm0.26}$ (+0.16%) | $\mathbf{13.37_{\pm0.29}}$ (-1.85%) | $15.46_{\pm0.71}$ (-7.42%) | $18.30_{\pm0.53}$ (-8.76%) | $21.40_{\pm0.66}$ (-10.75%) | $24.88_{\pm1.10}$ (-10.37%) | $\mathbf{30.43_{\pm1.07}}$ (-13.50%) |
| LAP-$\ell_2$ | $\mathbf{11.70_{\pm0.37}}$ (-2.21%) | $\mathbf{12.31_{\pm0.23}}$ (-1.90%) | $13.70_{\pm0.51}$ (+0.62%) | $\mathbf{15.42_{\pm0.62}}$ (-7.62%) | $\mathbf{17.94_{\pm0.91}}$ (-10.55%) | $\mathbf{21.38_{\pm1.24}}$ (-10.84%) | $\mathbf{24.36_{\pm1.55}}$ (-12.23%) | $30.55_{\pm3.04}$ (-13.16%) |

Table 18: Test error rates of VGG-19 on CIFAR-10 for channel pruning. Subscripts denote standard deviations, and bracketed numbers denote relative gains with respect to the best of MP-$\ell_1$ and MP-$\ell_2$. Unpruned models achieve 9.02% error rate.

|  | 34.30% | 28.70% | 24.01% | 20.09% | 16.81% | 14.06% | 11.76% | 9.84% |
|---|---|---|---|---|---|---|---|---|
| MP-$\ell_1$ | $9.25_{\pm0.23}$ | $9.81_{\pm0.36}$ | $10.12_{\pm0.15}$ | $10.77_{\pm0.73}$ | $14.28_{\pm1.57}$ | $14.53_{\pm1.48}$ | $18.84_{\pm3.53}$ | $23.71_{\pm4.94}$ |
| MP-$\ell_2$ | $9.40_{\pm0.23}$ | $9.73_{\pm0.52}$ | $10.27_{\pm0.18}$ | $10.61_{\pm0.74}$ | $12.26_{\pm1.79}$ | $13.74_{\pm1.96}$ | $17.70_{\pm3.46}$ | $33.27_{\pm15.72}$ |
| RP | $10.58_{\pm0.61}$ | $11.72_{\pm1.26}$ | $12.86_{\pm0.89}$ | $19.49_{\pm12.70}$ | $20.19_{\pm2.45}$ | $24.99_{\pm6.33}$ | $46.18_{\pm18.08}$ | $54.52_{\pm16.61}$ |
| LAP-$\ell_1$ | $\mathbf{9.05_{\pm0.23}}$ (-2.23%) | $9.46_{\pm0.25}$ (-2.75%) | $10.07_{\pm0.46}$ (-0.47%) | $\mathbf{10.53_{\pm0.27}}$ (-0.81%) | $10.95_{\pm0.19}$ (-10.73%) | $12.37_{\pm0.74}$ (-9.99%) | $15.50_{\pm0.81}$ (-12.43%) | $16.65_{\pm3.28}$ (-29.77%) |
| LAP-$\ell_2$ | $9.06_{\pm0.20}$ (-2.10%) | $\mathbf{9.42_{\pm0.36}}$ (-3.21%) | $\mathbf{9.74_{\pm0.37}}$ (-3.77%) | $\mathbf{10.53_{\pm0.40}}$ (-0.79%) | $\mathbf{10.74_{\pm0.22}}$ (-12.39%) | $\mathbf{11.87_{\pm0.33}}$ (-13.61%) | $\mathbf{13.51_{\pm0.27}}$ (-23.66%) | $\mathbf{15.67_{\pm2.78}}$ (-33.92%) |

# I  LOOKAHEAD FOR GLOBAL PRUNING

In this section, we present global pruning results for MP, LAP, OBD, OBD+LAP and LAP-act in Table 19 and Table 20. In this methods, we prune a fraction of weights with smallest scores (e.g. weight magnitude, lookahead distortion, Hessian-based scores) among all weights in the whole network. The suffix "-normalize" in the tables denotes that the score is normalized by the Frobenius norm of the corresponding layer's score. For MP, LAP, OBD+LAP and LAP-act, we only report the results for global pruning with normalization, as the normalized versions outperform the unnormalized ones. In the case of OBD, whose score is already globally designed, we report the results for both unnormalized and normalized versions.

As demonstrated in Section 3.2 for fixed layerwise pruning rates, we observe that LAP and its variants perform better than their global pruning baselines, i.e. MP-normalize and OBD. We also note that LAP-normalize performs better than MP with pre-specified layerwise pruning rates (appeared in Section 3.2), with a larger gap for higher levels of sparsity.

Table 19: Test error rates of FCN on MNIST for global pruning. Subscripts denote standard deviations, and bracketed numbers denote relative gains with respect to MP-normalize (for data-agnostic algorithms) and OBD-normalize (for data-dependent algorithms), respectively. Unpruned models achieve 1.98% error rate.

|  | 6.36% | 3.21% | 1.63% | 0.84% | 0.43% | 0.23% | 0.12% |
|---|---|---|---|---|---|---|---|
| MP-normalize (baseline) | $1.82_{\pm0.08}$ | $2.16_{\pm0.06}$ | $2.72_{\pm0.17}$ | $3.54_{\pm0.09}$ | $6.54_{\pm0.35}$ | $59.59_{\pm16.23}$ | $88.65_{\pm0.00}$ |
| LAP-normalize | $\mathbf{1.71}_{\pm0.09}$ | $\mathbf{2.07}_{\pm0.10}$ | $\mathbf{2.69}_{\pm0.09}$ | $\mathbf{3.42}_{\pm0.22}$ | $\mathbf{4.15}_{\pm0.07}$ | $\mathbf{6.68}_{\pm0.55}$ | $\mathbf{19.18}_{\pm3.81}$ |
|  | **(-6.16%)** | **(-4.26%)** | **(-1.03%)** | **(-3.33%)** | **(-36.57%)** | **(-88.79%)** | **(-78.36%)** |
| OBD (baseline) | $1.71_{\pm0.13}$ | $1.93_{\pm0.13}$ | $2.12_{\pm0.12}$ | $2.82_{\pm0.17}$ | $3.59_{\pm0.31}$ | $5.12_{\pm0.22}$ | $10.52_{\pm1.14}$ |
| OBD-normalize | $1.71_{\pm0.09}$ | $1.92_{\pm0.10}$ | $2.22_{\pm0.08}$ | $\mathbf{2.77}_{\pm0.25}$ | $3.55_{\pm0.19}$ | $4.99_{\pm0.26}$ | $11.08_{\pm2.73}$ |
|  | (-0.12%) | (-0.52%) | (+4.62%) | **(-1.84%)** | (-1.11%) | (-2.54%) | (+5.36%) |
| OBD+LAP-normalize | $1.84_{\pm0.13}$ | $2.00_{\pm0.13}$ | $2.22_{\pm0.16}$ | $2.93_{\pm0.34}$ | $3.55_{\pm0.27}$ | $5.04_{\pm0.76}$ | $\mathbf{8.33}_{\pm2.51}$ |
|  | (+7.48%) | (+3.73%) | (+4.91%) | (+3.97%) | (-1.22%) | (-1.52%) | **(-20.79%)** |
| LAP-act-normalize | $\mathbf{1.68}_{\pm0.13}$ | $\mathbf{1.80}_{\pm0.09}$ | $\mathbf{2.06}_{\pm0.10}$ | $2.80_{\pm0.19}$ | $\mathbf{3.50}_{\pm0.12}$ | $4.82_{\pm0.27}$ | $8.50_{\pm1.16}$ |
|  | **(-1.87%)** | **(-6.84%)** | **(-3.02%)** | (-0.78%) | **(-2.56%)** | **(-5.90%)** | (-19.21%) |

Table 20: Test error rates of Conv-6 on CIFAR-10 for global pruning. Subscripts denote standard deviations, and bracketed numbers denote relative gains with respect to MP-normalize (for data-agnostic algorithms) and OBD-normalize (for data-dependent algorithms), respectively. Unpruned models achieve 11.97% error rate.

|  | 10.62% | 8.86% | 7.39% | 6.18% | 5.17% | 4.32% | 3.62% |
|---|---|---|---|---|---|---|---|
| MP-normalize (baseline) | $12.42_{\pm0.17}$ | $13.14_{\pm0.35}$ | $14.17_{\pm0.40}$ | $15.39_{\pm0.40}$ | $17.57_{\pm0.46}$ | $21.04_{\pm0.42}$ | $24.40_{\pm1.57}$ |
| LAP-normalize | $\mathbf{11.81}_{\pm0.32}$ | $\mathbf{12.23}_{\pm0.25}$ | $\mathbf{12.44}_{\pm0.22}$ | $\mathbf{13.02}_{\pm0.12}$ | $\mathbf{13.73}_{\pm0.16}$ | $\mathbf{14.81}_{\pm0.34}$ | $\mathbf{15.97}_{\pm0.30}$ |
|  | **(-4.91%)** | **(-6.87%)** | **(-12.19%)** | **(-15.42%)** | **(-21.86%)** | **(-29.61%)** | **(-34.54%)** |
| OBD (baseline) | $12.03_{\pm0.64}$ | $12.30_{\pm0.53}$ | $12.64_{\pm0.15}$ | $13.16_{\pm0.23}$ | $13.75_{\pm0.45}$ | $14.70_{\pm0.53}$ | $16.11_{\pm0.50}$ |
| OBD-normalize | $\mathbf{11.69}_{\pm0.34}$ | $\mathbf{11.93}_{\pm0.21}$ | $12.58_{\pm0.08}$ | $\mathbf{12.87}_{\pm0.22}$ | $13.62_{\pm0.28}$ | $14.60_{\pm0.24}$ | $15.82_{\pm0.44}$ |
|  | **(-2.86%)** | **(-2.99%)** | (-0.47%) | **(-2.26%)** | (-0.89%) | (-0.67%) | (-1.75%) |
| OBD+LAP-normalize | $12.11_{\pm0.32}$ | $12.66_{\pm0.46}$ | $13.36_{\pm0.47}$ | $13.60_{\pm0.33}$ | $14.05_{\pm0.34}$ | $14.98_{\pm0.33}$ | $15.82_{\pm0.39}$ |
|  | (+0.68%) | (+2.96%) | (+5.66%) | (+3.30%) | (+2.24%) | (+1.89%) | (-1.80%) |
| LAP-act-normalize | $11.92_{\pm0.23}$ | $12.24_{\pm0.05}$ | $\mathbf{12.51}_{\pm0.45}$ | $12.89_{\pm0.36}$ | $\mathbf{13.53}_{\pm0.41}$ | $\mathbf{14.21}_{\pm0.40}$ | $\mathbf{15.42}_{\pm0.16}$ |
|  | (-0.90%) | (-0.49%) | **(-1.08%)** | (-2.05%) | **(-1.54%)** | **(-3.31%)** | **(-4.26%)** |

## J    LAP-ALL: LOOKING AHEAD THE WHOLE NETWORK

We also report some experimental results on a variant of lookahead pruning, coined LAP-all, which treats (a linearized version of) the whole network as an operator block. More specifically, one attempts to minimize the Frobenius distortion of the operator block

$$\min_{M_i:\|M_i\|_0=s_i} \|\mathcal{J}_{d:i+1}\mathcal{J}(W_i)\mathcal{J}_{i-1:1} - \mathcal{J}_{d:i+1}\mathcal{J}(M_i \odot W_i)\mathcal{J}_{i-1:1}\|_F\,,$$

where $\mathcal{J}_{i+j:i} := \mathcal{J}(W_{i+j})\mathcal{J}(W_{i+j-1})\cdots\mathcal{J}(W_i)$.

We test LAP-all on FCN under the same setup as in Section 3.2, and report the results in Table 21. All figures are averaged over five trials.

We observe that LAP-all achieves a similar level of performance to LAP, while LAP-all underperforms under a high-sparsity regime. We suspect that such shortfall originates from the accumulation of error terms incurred by ignoring the effect of activation functions, by which the benefits of looking further fades. An in-depth theoretical analysis for the determination of an optimal "sight range" of LAP would be an interesting future direction.

Table 21: Test error rates of FCN on MNIST, with LAP-all variant. Subscripts denote standard deviations. Unpruned models achieve 1.98% error rate.

|  | 6.36% | 3.21% | 1.63% | 0.84% | 0.43% | 0.23% | 0.12% |
|---|---|---|---|---|---|---|---|
| MP (baseline) | $1.75_{\pm 0.11}$ | $2.11_{\pm 0.14}$ | $2.53_{\pm 0.09}$ | $3.32_{\pm 0.27}$ | $4.77_{\pm 0.22}$ | $19.85_{\pm 8.67}$ | $67.62_{\pm 9.91}$ |
| RP | $2.36_{\pm 0.13}$ | $2.72_{\pm 0.16}$ | $3.64_{\pm 0.17}$ | $17.54_{\pm 7.07}$ | $82.48_{\pm 4.03}$ | $88.65_{\pm 0.00}$ | $88.65_{\pm 0.00}$ |
| LAP | $1.67_{\pm 0.11}$ | $\mathbf{1.89}_{\pm \mathbf{0.12}}$ | $\mathbf{2.48}_{\pm \mathbf{0.13}}$ | $3.29_{\pm 0.06}$ | $\mathbf{3.93}_{\pm \mathbf{0.26}}$ | $\mathbf{6.72}_{\pm \mathbf{0.44}}$ | $\mathbf{16.45}_{\pm \mathbf{5.61}}$ |
| LAP-all | $\mathbf{1.64}_{\pm \mathbf{0.05}}$ | $2.06_{\pm 0.17}$ | $2.53_{\pm 0.15}$ | $\mathbf{3.23}_{\pm \mathbf{0.13}}$ | $4.01_{\pm 0.10}$ | $6.78_{\pm 0.44}$ | $25.64_{\pm 5.42}$ |

## K    COMPARISON WITH SMALLER NETWORKS

As a sanity check, we compare the performance of large neural networks pruned via MP and LAP to the performance of a small network. In particular, we prune VGG-16, VGG-19, and ResNet-18 trained on CIFAR-10 dataset, to have a similar number of parameters to MobileNetV2 (Sandler et al., 2018). For training and pruning VGGs and ResNet, we follows the prior setup in Appendix A while we use the same setup for training MobileNetV2 (Adam optimizer with learning rate of $3 \cdot 10^{-4}$ with batch size 60, and trained 60k steps). We observe that models pruned via LAP (and MP) exhibit better performance compared to MobileNetV2, even when pruned to have a smaller number of parameters.

Table 22: Test error rates of various networks on CIFAR-10. Subscripts denote standard deviations, and bracketed numbers denote relative gains with respect to the unpruned MobileNetV2.

|  | VGG-16 | VGG-19 | ResNet-18 | MobileNetV2 |
|---|---|---|---|---|
| Unpruned | $9.33_{\pm 0.15}$ | $9.02_{\pm 0.36}$ | $8.68_{\pm 0.21}$ | $9.81_{\pm 0.30}$ |
| MP | $8.92_{\pm 0.18}$ (-9.07%) | $9.46_{\pm 0.25}$ (-3.57%) | $\mathbf{7.70}_{\pm 0.23}$ $\mathbf{(-21.51\%)}$ | - |
| LAP | $\mathbf{8.77}_{\pm 0.20}$ $\mathbf{(-10.60\%)}$ | $\mathbf{9.30}_{\pm 0.25}$ $\mathbf{(-5.20\%)}$ | $7.73_{\pm 0.29}$ (-21.20%) | - |
| # weight parameters | 2.09M/14.72M (14.23%) | 2.06M/20.03M (10.28%) | 2.17M/10.99M (19.17%) | 2.20M |

## L    WHERE IS THE PERFORMANCE GAIN OF LAP COMING FROM?

In this section, we briefly discuss where the benefits of the sub-network discovered by LAP comes from; does LAP subnetwork have a better generalizability or expressibility? For this purpose, we look into the generalization gap, i.e., the gap between the training and test accuracies, of the hypothesis learned via LAP procedure. Below we present a plot of test accuracies (Fig. 4a) and a plot of generalization gap (Fig. 4b) for FCN trained with MNIST dataset. The plot hints us that the network structure learned by LAP may not necessarily have a smaller generalizability. Remarkably, the generalization gap of the MP-pruned models and the LAP-pruned models are very similar to each other; the benefits of LAP subnetwork compared to MP would be that it can express a better-performing architecture with a network of similar sparsity and generalizability.

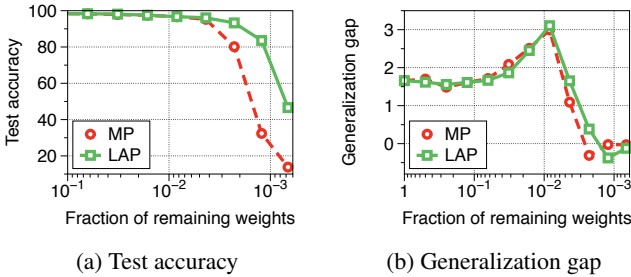

(a) Test accuracy                   (b) Generalization gap

Figure 4: Test accuracy and generalization gap of FCN trained on MNIST.

## M    CONNECTIONS TO IMPLICIT BIAS OF SGD

Another theoretical justification of using the lookahead distortion (Eq. (5)) for neural networks with nonlinear activation functions comes from recent discoveries regarding the implicit bias imposed by training procedures using stochastic gradient descent. More specifically, Du et al. (2018) proves the following result, generalizing the findings of Arora et al. (2018): For any two neighboring layers of fully-connected neural network using positive homogeneous activation functions, the quantity

$$\|W_{i+1}[:,j]\|_2^2 - \|W_i[j,:]\|_2^2 \tag{26}$$

remains constant for any hidden neuron $j$ over training via gradient flow. In other words, the total outward flow of weights is tied to the inward flow of weights for each neuron. This observation hints at the possibility of a relative undergrowth of weight magnitude of an 'important' connection, in the case where the connection shares the same input/output neuron with other 'important' connections. From this viewpoint, the multiplicative factors in Eq. (5) take into account the abstract notion of neuronal importance score, assigning significance to connections to the neuron through which more gradient signals have flowed through. Without considering such factors, LAP reduces to the ordinary magnitude-based pruning.

