# OpenReview forum: "Lookahead: A Far-sighted Alternative of Magnitude-based Pruning"
_ICLR.cc/2020/Conference — Accept (Poster)_

### Official Review · AnonReviewer3 · 2019-10-11
**Official Blind Review #3**

**Rating:** 6

**Review:**

This paper proposed Lookahead Pruning (LAP), a new method for model weight pruning to generate neural networks with sparse weights. The authors interpret the conventional magnitude pruning (MP) as pruning each layer individually while the proposed lookahead method considers neighboring layers' weights. Specifically, the proposed methods prune weights which introduce small distortion to the Jacobian matrix of 3 consecutively connected (linear) layers; and the conventional magnitude pruning can be viewed as a degenerated/special case of the LAP. The primary contributions of the paper are: 1) the authors propose the LAP method for fully connected layers; 2) they present empirical applications/extensions to models with non-linear-activations and batchnorms (which are two important components of modern neural networks); 3) The authors empirically show that LAP (and its variants such LAP-forward and LAP-forward-seq) can generate sparse models with better test accuracy than MP, across fully-connected network (FCN), and Conv models (such as VGG and ResNet) on MNIST and CIFAR dataset.

I think the method in this paper is well motivated both mathematically (minimizing distortions of Jacobian) and intuitively (considering multiple layers holistically). The empirical benefits of the proposed method is properly validated against MP in various dataset and models. Also the paper is well written. Thus I give weak accept and I am willing to raise the score if convincing clarification on the following questions can be provided in author responses and in the future draft:

1. As the LAP method introduces higher computation overhead in pruning weights, I was wondering how it compares to MP in terms of run-time-efficiency. As LAP requires computing a score for each single weight value (though some of the computationally heavy terms can be reused), it is important to discuss how long does LAP pruning take, comparing to the run-time of retraining (after pruning). This will help further evaluate the empirical efficiency of the LAP method.

2. The experiment focus on CNNs while the authors advocating versatility of the methods. Thus I was wondering how LAP performs on models in other domains, such as transformer-based NLP models.

3. (Relatively minor). The experiment purely focused on comparing pruning methods. To demonstrate the empirical merits of LAP, I think it will be more convincing to also compare with naive baselines of using narrow / shallower networks such as in Sohoni et al.[1]. This will demonstrate that pruning itself and LAP as an instantiation of pruning should be considered over these naive narrow / shallower baselines with the same amount of weight parameters as pruned models.

4. The tables (table 1-5) are massive but the take-away message is not crystal clear in the text or captions of the table. Is the take-away message something like 1) one might want to use LAP-forward(backward) over LAP when you have very high sparsity, and 2) the sequential versions can further enhance the performance?


Minor suggestions to improve the paper:

1. Line 5 and 6 in Algorithm 1 is confusing. I suppose the authors mean selecting the weights which trigger small value for L to zero-out. To me the current line 5,6 does not directly reflect this.

2. It might be good to provide a proof in appendix on equation 5), it take me quite a few minutes to verify it. Providing a proof can help readers to read more smoothly.

3. The order of the content can be slight reorganized, e.g. why talking about the adaptation of LAP on batchnorm after you discussed all the directional and sequential variants of LAP?

Reference
[1] Low-Memory Neural Network Training: A Technical Report. Sohoni et al.



**Experience Assessment:**

I have read many papers in this area.

**Review Assessment: Checking Correctness Of Derivations And Theory:**

I carefully checked the derivations and theory.

**Review Assessment: Checking Correctness Of Experiments:**

I carefully checked the experiments.

**Review Assessment: Thoroughness In Paper Reading:**

I read the paper thoroughly.

---

> ### Author Response · Authors · 2019-11-13
> **Response to R3**
>
> We sincerely appreciate your valuable comments, efforts and time. We are grateful for all positive comments: easy to implement (by R2 and R5), good empirical performance (by R3 and R5), good write-up (by you, R2 and R4) and novelty (by R4). In the revised manuscript, we have updated or newly added (Section 2, Section 3, Appendix C, E, F, G, H, I, J, K) according to the reviewers’ comments and colored them blue. We address each comment in detail, one by one as below.
>
> # Major Suggestions
> (Q1) Computational complexity of LAP.---------------------------------------------------------------------
>
> (A1) We thank the reviewer for pointing this out; this is indeed an important issue to be addressed. As the reviewer has correctly indicated, most computationally heavy terms can be reused; indeed, the computation of the lookahead distortion can be done tensor-wise instead of computing separately for each connection.
>
> To persuade the readers further, we have recorded the run-time of pruning operations for models appearing in the submission. Pruning VGG-19 with Intel Xeon-E5-2630v4@2.20GHz processor, MP takes approximately 0.9 seconds and LAP takes approximately 1.2 seconds, which is negligible compared to the retraining time. More generally, we observe that the computational overhead from computing the lookahead distortion is less than 10% of the computational cost of MP. More computations are required to handle batch normalization layers, but the added computing time did not exceed 50%.
>
> The results, with a more detailed explanation, have been added to Appendix G of the revised manuscript.
>
> (Q2) Experiments on other domains.-------------------------------------------------------------------------
>
> (A2) Due to requests from reviewers, most of added experiments focus on a more rigorous assessment of LAP on larger datasets and models for computer vision tasks. We would continue to add more experiments on different tasks, as time permits.
>
> (Q3) Narrower/Shallow net baseline.-------------------------------------------------------------------------
>
> (A3) We compared the CIFAR-10 classification performance of VGG-16, VGG-19, ResNet-18 pruned MP/LAP, with the performance of MobileNetV2 [1], which is a small sized network having only 2.2M number of parameters. We observe that pruned models exhibit better performance compared to MobileNetV2 under the smaller number of parameters. The detailed results are summarized in Appendix K of the revised manuscript.
>
> (Q4) Takeaway message from the tables.-------------------------------------------------------------------
>
> (A4) Thank you for pointing this out. The main messages that we intended to deliver are as following.
> Looking ahead a layer helps improving the accuracy of the pruned models, over MP.
> Sequential methods help stabilizing the performance of LAP, which helps improving the performance at extreme sparsity levels. For larger models with batch-norm, however, this advantage is not always present.
> Following the reviewer’s suggestion, we have updated texts of Section 3 to make this point clearer.
>
> ------------------------------------------------------------------------------------------------------------------------------------------------
>
> # Minor suggestions.
> We thank the reviewer for making various suggestions to improve our manuscript. We have updated accordingly.
>
> ------------------------------------------------------------------------------------------------------------------------------------------------
>
> [1] Sandler et al., “MobileNetV2, Inverted Residuals and Linear Bottlenecks”, CVPR 2018

---

> > ### Comment · AnonReviewer3 · 2019-11-14
> > **Thanks for addressing my comments**
> >
> > Dear authors,
> >
> > Thanks for the efforts in providing additional results and discussions to address my comments.
> >
> > I appreciate the efforts in demonstrating the efficiency of the method and the comparison to more efficient baselines.
> >
> > I will keep the rating for now. If the authors have the time to present convincing results on non-vision tasks (e.g. NLP). I will raise the score to 7.
> >
> > Thanks.

---

> > > ### Author Response · Authors · 2019-11-15
> > > **Thanks for your response!**
> > >
> > > Dear reviewer,
> > >
> > > Thank you for your continuing effort to provide constructive feedback until the end of the response/discussion phase.
> > >
> > > To best respond to your feedback, we will continue our effort to include non-vision experimental results in our final manuscript, with the same level of rigor (5x duplication) as done in vision experiments.
> > >
> > > Thanks,
> > > Authors.

---

### Official Review · AnonReviewer4 · 2019-11-01
**Official Blind Review #4**

**Rating:** 6

**Review:**

This paper proposes a new magnitude-based pruning method (and a few variants) by extending the single-layer distortion minimization problem to multi-layer cases so that the correlation between layers is taken into account. Particularly, the authors take into account the weight tensors of neighboring layers in addition to the original layer. The proposed algorithm looks promising and interesting. Empirically, the authors show that the proposed method consistently outperforms the standard magnitude-based pruning method.

Overall, the paper is well-written and I think the algorithm is novel. Therefore, I've given the score of 6.

Comments:
(1) It seems obvious that the proposed method would increase the computation cost, but the authors didn't give any discussion or results on that.
(2) Although the main focus of the paper is magnitude-based pruning, I think the authors should include one baseline of Hessian-based pruning methods for comparison. As I know, the computation overhead of Hessian-based methods (e.g., OBD) is relatively small for the networks used in this paper. In particular, Hessian-based pruning methods can also be interpreted as a distortion minimization problem but in a different space/metric. So I wonder if the authors can extend LAP to Hessian-based pruning methods.
(3) The authors introduced LAP with deep linear networks. However, the details of LAP in non-linear network are missing. I encourage the authors to fill in the details in section 2.2 in the next revision.
(4) Currently, all experiments are done on CIFAR-10 dataset. I wonder if the author can include one more dataset. For example, comparison between MP and LAP on Tiny-ImageNet or even ImageNet. As I know, experiments of ImageNet can fit into a 4-GPU server for magnitude-based methods.

**Experience Assessment:**

I have published one or two papers in this area.

**Review Assessment: Checking Correctness Of Derivations And Theory:**

I assessed the sensibility of the derivations and theory.

**Review Assessment: Checking Correctness Of Experiments:**

I assessed the sensibility of the experiments.

**Review Assessment: Thoroughness In Paper Reading:**

I read the paper at least twice and used my best judgement in assessing the paper.

---

> ### Author Response · Authors · 2019-11-13
> **Response to R4**
>
> We sincerely appreciate your valuable comments, efforts and time. We are grateful for all positive comments: easy to implement (by R2 and R5), good empirical performance (by R3 and R5), good write-up (by you, R2 and R3) and novelty (by you). In the revised manuscript, we have updated or newly added (Section 2, Section 3, Appendix C, E, F, G, H, I, J, K) according to the reviewers’ comments and colored them blue. We address each comment in detail, one by one as below.
>
> (Q1) Comparison of computational cost-----------------------------------------------------------------
>
> (A1) We thank the reviewer for pointing this out. As the reviewer noted, LAP requires more computation than MP. To confirm our initial claim that the overhead is not prohibitively significant, we ran and timed 100 trials of MP and LAP on some neural network models appearing in the manuscript. It turns out that the computation of lookahead distortion introduced only less than 10% overhead over magnitude pruning, with the help of popular tensor-handling frameworks. Handling batch normalization layers required an additional computations, but the overhead did not exceed 50% in this case as well.
>
> The results, with a more detailed explanation, have been added to Appendix G of the revised manuscript.
>
> (Q2) OBD / Lookahead for OBD-----------------------------------------------------------------------------
>
> (A2) Thank you for suggesting a comparison to OBD. In our revised manuscript, we now
> 1) distinguish the data-agnostic pruning schemes (MP and LAP) from data-dependent schemes (e.g. OBD) more explicitly (in revised introduction and section 2),
> 2) provide two data-dependent variants of LAP; one based on your suggestion of looking ahead with Hessian-based scores (coined OBD+LAP) and another based on the “activation probability” described in the previous manuscript (in a newly added Appendix G), and
> 3) empirically evaluate two variants and compare their performance/computation with optimal brain damage, implemented with recently released “BackPACK” package [1] (detailed in Appendix G).
>
> As the reviewer expected, we observed that OBD+LAP variant performing better compared to OBD. Indeed, we also report that another variant based on activation probability works even better, without having to calculate Hessian diagonals via back-prop.
>
> (Q3) LAP in non-linear networks.---------------------------------------------------------------------------
>
> (A3) We agree with the reviewer’s concern that the fact that we use the same algorithm for deep linear networks and deep nonlinear networks, was not crystal clear in our previous version of the manuscript. Following the reviewer’s suggestion, we have added more detailed explanations in Section 2.1 (Section 2.2. has been merged to Section 2.1, with sending some less relevant materials to appendices).
>
> (Q4) More experiments on different datasets.----------------------------------------------------------
>
> (A4) The scalability of our algorithm is indeed an important issue. We have conducted additional experiments on Tiny-ImageNet (also averaged over 5 trials), and report the results in Section 3.3.
>
> We would continue to add more experimental results, as time permits.
>
> -------------------------------------------------------------------------------------------------------------------------
>
> [1] Anonymous, “BackPACK: packing more into backprop,” under review for ICLR 2020.

---

> > ### Comment · AnonReviewer4 · 2019-11-13
> > **Thank you for addressing my concerns.**
> >
> > I've read your response and the updated paper. I'm quite satisfied with your response.
> >
> > I recommend to accept the paper though I keep my original rating (if there's an option of 7, I will definitely increase my rating).

---

> > > ### Author Response · Authors · 2019-11-15
> > > **Thanks for your response!**
> > >
> > > Dear reviewer,
> > >
> > > We are happy to hear that our response was satisfactory for you. We also appreciate your valuable time and efforts to help us improve our manuscript.
> > >
> > > Thanks,
> > > Authors.

---

### Official Review · AnonReviewer2 · 2019-11-01
**Official Blind Review #2**

**Rating:** 6

**Review:**

*Summary*
The paper proposes a multi-layer alternative to magnitude-based pruning. The operations entailed in the previous, current, and subsequent layers are treated as linear operations (by omitting any nonlinearities), weights are selected for pruning to minimize the "Frobenius distortion", the Frobenius norm of the difference between products of the (i-1, i, i+1)-layer Jacobians with and without the selected weight. This simplifies to a cost-effective pruning criterion. In spite of the simplistic linear setting assumed for the derivation, results show the criterion prunes better than weight-based methods at unstructured pruning of a variety of modern architectures with CIFAR-10, particularly excelling at higher sparsity.

*Rating*
The paper has some clear positives, particularly:
    + Clear writing and formatting
    + Simple method
    + Good structure for the experimental analysis (with 5x replications!)
However there are a few limitations, noted below; while none is fatal on its own, in total the limitations have led me to recommend "weak reject" currently.

Limitations of the method:
(1) Residual networks: The lack of an explicit strategy for handling residual connections (and the accompanying worsened relative performance) is a notable limitation since residual/skip connections are nearly universal in state-of-the-art large networks. The performance was shown to still be *slightly* better than with magnitude pruning.
(2) Global ranking: Since connections are pruned layerwise, rather than taking the best-k neurons across the entire network at once, I assume that the LAP pruning criterion doesn't scale reasonably across layers. This implies that the method cannot be used to learn network structure. Instead the user must decide the desired number of neurons at each layer.
(3) Structured pruning: There is no mention of pruning entire convolutional kernels or "neurons" at once, so I assume that only individual weights were pruned. Since structured pruning is the simplest way to achieve speedup in network inference (as opposed to merely reduction in model size), how does the LAP criterion perform when adapted for structured pruning, e.g. removing filters/neurons with the best average LAP score?

Limitations of the experiments:
(4) Baselines: While the paper is explicitly focused on an easy to compute replacement for magnitude-based pruning, there are a wide variety of alternative methods available. These vary in complexity, runtime, etc., but they deserve mention and either explicit comparison in the experiments or reasoning to justify the omission of such comparisons.
(5) ImageNet: (Insert obligatory statement about the ubiquity of ImageNet experiments, ...) While it is cliche to request ImageNet experiments and CIFAR-10 is a helpful stand-in, they would be really nice to have.
(6) Activations after non-linearities: While Fig. 3 and the remaining experiments present a reasonable case that the presence of non-linearities doesn't prevent LAP from improving upon magnitude-based pruning, it doesn't resolve the issue either. Whether considering negative values clipped by with ReLU or large magnitude values that are squashed by sigmoid and tanh, the linear-only model is a poor approximation for some unknown fraction of neurons for probably most inputs. Does this mean that LAP is underperforming in those cases? Are those cases sufficiently rare or randomly distributed that they are merely noise? Is there another mechanism at play? In practical terms, how much does the activation rate (positive for ReLU, linear/unsquashed for sigmoid/tanh) vary by neuron? This seems like a reasonably simple thing to compute and incorporate into pruning.

*Notes*
Eq. (4): Is (4) simply the one-step/greedy approximation to the optimization in (3)? If so, it may be helpful to state this explicitly. Also, is $w = W_i[j,k]$? If so, this is useful to explicitly state.
Sec 2.1: Consider noting that the linear-model setup is used to construct the method, but non-linearities are addressed subsequently
Sec 2.2: Is the activation probability p_j used in practice, or is it merely an explanatory device?
pg5: "gradually prune (p/5)%" and marked with a suffix '-seq'"
pg5: note that residual connections are discussed in the experiments?
Tables 3-6: note that these all use CIFAR-10

**Experience Assessment:**

I have published one or two papers in this area.

**Review Assessment: Checking Correctness Of Derivations And Theory:**

I assessed the sensibility of the derivations and theory.

**Review Assessment: Checking Correctness Of Experiments:**

I carefully checked the experiments.

**Review Assessment: Thoroughness In Paper Reading:**

I read the paper thoroughly.

---

> ### Author Response · Authors · 2019-11-13
> **Response to R2 (1/2)**
>
> We sincerely appreciate your valuable comments, efforts and time. We are grateful for all positive comments: easy to implement (by you and R5), good empirical performance (by R3 and R5), good write-up (by you, R3 and R4) and novelty (by R4). In the revised manuscript, we have updated or newly added (Section 2, Section 3, Appendix C, E, F, G, H, I, J, K) according to the reviewers’ comments and colored them blue. We address each comment in detail, one by one as below.
>
> Below, we respond to some of the questions/concerns raised by the reviewer.
>
> (Q1) Residual networks----------------------------------------------------------------------------------------
>
> (A1) We deeply respect your concern about residual connections. On the other hand, we would like to emphasize that LAP still makes over 15% relative accuracy gain over MP on ResNet-18 (Table 4, at 0.36% sparsity level), even without adding complicated mechanisms to account for residual connections.
>
> Also, we note that more experimental results with ResNet and WRN trained on Tiny-ImageNet data have been added in Table 6-7, 11-12 of the revised manuscript, where LAP consistently outperforms MP.
>
> (Q6) Activations after non-linearities----------------------------------------------------------------------
>
> (A6) To better respond to this question, we have followed the reviewer’s suggestion to implement the extension of LAP suggested in Section 2.2. based on the activation probability of the neurons. As an estimation of the activation probability requires an additional use of training dataset, we also implemented optimal brain damage [1] for a fairer comparison.
> The algorithm and related experiments are explained in Appendix F of the revised manuscript.
>
> We observe that accounting for nonlinearities of ReLU dramatically improves the performance of LAP methods to provide a better accuracy than the OBD baseline. On the other hand, we note that “estimating the nonlinearities” require additional knowledge (and computations) about the data domain beside the trained model, as OBD does. Conversely, assuming linearity can be thought of as accounting for a lack of knowledge about the data. To make this point clear, we revised Section 2.2.
>
> (Q4) Baselines-----------------------------------------------------------------------------------------------------
>
> (A4) As the reviewer indicated correctly, our primary focus was to provide a better understanding on the magnitude-based pruning methods (which is known to show performance comparable to state-of-the-art methods), by taking a principled perspective toward MP to deduce a better alternative. We have revised introduction and section 2 to make this point clearer.
>
> In addition, we have added Appendix F to the revised manuscript devoted to a discussion of LAP under the setup where training data is available (as described in the response of 6), and made explicit experimental comparisons with optimal brain damage [1].
>
> ---------------------------------------------------------------------------------------------------------------------------------
>
> [1] LeCun et al., “Optimal brain damage,” NIPS 1989.

---

> > ### Author Response · Authors · 2019-11-13
> > **Response to R2 (2/2)**
> >
> > (Q2) Global ranking----------------------------------------------------------------------------------------------
> >
> > (A2) In the initial manuscript, the main reason for using the fixed pruning ratio (identical to the setup in Frankle and Carbin [2] except for FCN) was to ensure fairness in comparing MP with LAP. However, to resolve the reviewer’s concern, we additionally implemented and tested global pruning algorithms based on LAP. The experimental results suggest that LAP could indeed be extended to global pruning (Appendix I of the revised manuscript). Our global pruning schemes outperform MP and optimal brain damage (OBD) [1] baselines, respectively, while OBD score is already computed non-locally. In addition, we note that further tuning of layerwise pruning ratio could improve the performance of LAP and its variants.
> >
> > (Q3) Structured pruning----------------------------------------------------------------------------------------
> >
> > (A3) We thank the reviewer for pointing out the possibility of using LAP criterion for structured pruning. As the reviewer noted, structured pruning is known to be an effective strategy to provide a speedup in network inference.
> >
> > Following the reviewer’s suggestion, we have conducted channel pruning experiments using LAP criterion to replace the magnitude criterion. It turns out that LAP-based channel pruning also outperforms the naïve magnitude baseline, coherent to our findings in unstructured pruning. The detailed results can be found in Appendix H of the revised manuscript.
> >
> > (Q5) ImageNet-----------------------------------------------------------------------------------------------------
> >
> > (A5) Scalability to bigger datasets is certainly an important issue to be addressed. As the reviewer suggested, we have conducted additional experiments (with five independent trials) with Tiny-ImageNet dataset to confirm the benefits of LAP over MP once again. The results are added to Section 3.3.
> >
> > We would continue to add more experiments under various setups, as time permits.
> >
> > ---------------------------------------------------------------------------------------------------------------------------
> > +) Thank you for making constructive suggestions as a note, which helped us improve the manuscript.
> >
> > [1] LeCun et al., “Optimal brain damage,” NIPS 1989.
> > [2] Frankle and Carbin, “The lottery ticket hypothesis: finding sparse, trainable neural networks,” ICLR 2019.

---

### Official Review · AnonReviewer5 · 2019-11-03
**Official Blind Review #5**

**Rating:** 6

**Review:**

[Summary]:
This paper interprets the underlying objective of magnitude pruning(MP) as minimizing the Frobenius distortion of a single layer. Then the authors provide a motivating example to show that MP may cause a large Frobenius distortion due to ignoring the inter-layer interactions. Based on this observation, the authors propose a simple modification to MP by explicitly enforcing to minimize the Frobenius distortion of an operator block consisting of multiple linear layers, and demonstrate better performance than MP on CIFAR10 and MNIST datasets.

[Pros]:
- The proposed algorithm is simple and easy to implement.
- The empirical results show that the proposed method beat MP consistently, and in particular for high sparsities.
- The ablation study about LAP, LFP and LBP is interesting.

[Cons & Questions]:
(1) Minimizing Frobenius distortion is not meaningful, and it only minimizes the change in the output to the first order. Moreover, I don’t think minimizing the change in the output is as meaningful as minimizing the increase in training error as is done in Hessian-based pruning methods, e.g., Optimal Brain Damage (OBD) ( LeCun et al. 1989). My reason is that it is possible that the output changes a lot, but the training error still remains low after pruning.

(2) Can the authors elaborate what are the advantages of MP/LAP over Hessian-based pruning, such as OBD? OBD only needs the diagonal Hessian matrix and is also tractable, and MP is only a special case of OBD when the Hessian is an identity matrix. I am not quite convinced MP can achieve state-of-the-art performance, and also Gale et al. (2019) did not include any Hessian-based pruning algorithm into comparisons. Therefore, it would be great if the authors can provide more justifications for why MP/LAP is advantageous to Hessian-based pruning methods, e.g., OBD. Besides, I would be happy to see the authors can include OBD as a baseline in the experiments.

(3) The interpretation of the objective of MP as minimizing the Frobenius distortion is well-known, and more general results are already presented in Dong et al. (2017). The authors should discuss it in the main paragraph and give the corresponding credits to Dong et al. (2017).

(4) Why do you need to specify the pruning ratio for each layer manually? It makes MP or LAP hard to use in practice, and it usually needs expert knowledge to specify the pruning ratios for different layers. For Hessian-based methods, it reflects the change in loss and thus can be used to automatically determine the pruning ratio at each layer.

(5) In table 1 and table 2, LFP is better than LBP when pruning ratio is low, while LBP becomes better for high pruning ratios. Is there any explanations?

(6) My understanding of the LAP is that it tries to minimize the Frobenius distortion of the input-output Jacobian matrix of the operator block. In the paper, the operator block consists of 3 consecutive linear layers. I am curious about what is the performance if we treat the entire network as an operator block?

(7) The experiments are only conducted on MNIST and CIFAR-10, which are overly simple. Further experiments on larger datasets will make the paper stronger and the results more convincing. Anyway, this is not a big issue, but I encourage the authors can test the proposed method on more challenging datasets and make fair comparisons.

Overall, my rating is largely due to the concerns of (1)&(2).

[References]:
Y. LeCun, J. S. Denker, and S. A. Solla. Optimal brain damage. In Advances in Neural Information Processing Systems, 1989.
T. Gale, E. Elsen, and S. Hooker. The state of sparsity in deep neural networks. arXiv preprint 1902.09574, 2019.
Dong, Xin, Shangyu Chen, and Sinno Pan. "Learning to prune deep neural networks via layer-wise optimal brain surgeon." Advances in Neural Information Processing Systems. 2017.


**Experience Assessment:**

I have published one or two papers in this area.

**Review Assessment: Checking Correctness Of Derivations And Theory:**

I carefully checked the derivations and theory.

**Review Assessment: Checking Correctness Of Experiments:**

I carefully checked the experiments.

**Review Assessment: Thoroughness In Paper Reading:**

I read the paper thoroughly.

---

> ### Author Response · Authors · 2019-11-13
> **Response to R5 (1/2)**
>
> We sincerely appreciate your valuable comments, efforts and time. We are grateful for all positive comments: easy to implement (by you and R2), good empirical performance (by you and R3), good write-up (by R2, R3 and R4) and novelty (by R4). In the revised manuscript, we have updated or newly added (Section 2, Section 3, Appendix C, E, F, G, H, I, J, K) according to the reviewers’ comments and colored them blue. We address each comment in detail, one by one as below.
>
> (Q1) Magnitude-based methods vs. Hessian-based methods.----------------------------------------------
>
> (A1) We agree with the reviewer’s point: Hessian-based methods provide a more direct analysis of loss, while our approach based on Frobenius distortion could only provide an upper bound on the loss by distortion.
>
> On the other hand, a distinguishing property of MP/LAP is its data-agnostic nature. Indeed, distortion-based approach could be interpreted minimizing a worst-case loss without any knowledge on the training data.
>
> Such data-agnostic approaches have the following clear advantages.
> - Ease of computation: While OBD is known to be efficiently computable via back-propagation, magnitude-based methods are much faster in general. We have implemented OBD and recorded its runtime in the newly added Table 15. Comparing to the test time of MP/LAP (presented in the newly added Table 16), OBD requires more than x200 computation time, even with using a recently introduced “BackPACK” package [1] designed for an efficient computation of Hessian diagonal (and using additional GPU).
> - Flexibility: Data-agnostic approaches can be flexibly used in the problem setups where we do not have additional access to the data that was used to train the model. For instance, Morcos et al. [2] recently studied a “transfer” of subnetwork discovered by MP to a relevant dataset, followed by training in the target domain. Given a trained model from a source domain only, data-agnostic methods (including LAP) can be applied for such tasks without having to access a source domain dataset, unlike their Hessian-based counterparts.
>
> (Q2) Evaluation with OBD--------------------------------------------------------------------------------------------
>
> (A2) We strongly agree with the reviewer’s concern that recent papers claiming the near-optimality of magnitude-based methods (usually via dynamic reconnection methods of Zhu and Gupta [3]) often lack a direct comparison to OBD.
>
> To this end, we have implemented and tested OBD on FCN and Conv-6 in Appendix F of the revised manuscript. From the experiments, we make the following observations.
> - As LeCun et al. [4] claimed, naïve magnitude-based pruning underperforms OBD.
> - Somewhat surprisingly, LAP can be refined to outperform OBD by using the training dataset. Drawing inspiration from the comments of Reviewer#2, we designed an LAP variant taking into account the activation probabilities (a.k.a. APoZ [5]) of each neuron, as previously described in Section 2.2 of the initial manuscript. This variant provides better test accuracies than OBD, especially in the high-sparsity regime. The details are presented in Appendix F of the revised manuscript.
>
> Finally, we emphasize that our study on LAP is aimed toward a better understanding of why magnitude-based, data-agnostic pruning methods perform sufficiently well, instead of claiming that it is state-of-the-art. To make this point clearer, we have toned down the descriptions appearing in the introduction.
>
>
> ------------------------------------------------------------------------------------------------------------------------------------
>
>
> [1] Anonymous, “BackPACK: packing more into backprop,” under review for ICLR 2020.
> [2] Morcos et al., “One ticket to win them all: generalizing lottery ticket initializations across datasets and optimizers,” to appear in NeurIPS 2019.
> [3] Zhu and Gupta, “To prune or not to prune: exploring the efficacy of pruning for model compression,” ICLR 2018 workshop.
> [4] LeCun et al., “Optimal brain damage,” NIPS 1989.
> [5] Hu et al. “Network trimming: a data-driven neuron pruning approach towards efficient deep architectures,” arXiv 2016.

---

> > ### Author Response · Authors · 2019-11-13
> > **Response to R5 (2/2)**
> >
> > (Q3) Credit to Dong et al. (2017)-----------------------------------------------------------------------------
> >
> > (A3) We thank the reviewer for noting the related work of Dong et al. [6]. We fully agree that the work should be highly accredited for being the first (up to our knowledge) to highlight the role of Frobenius norm of Jacobian matrices for distortion analysis. Still, we remark that our approach steps further by explicitly considering inter-layer dependencies to derive a novel scoring method. We have added a reference to [6] in Section 2.
> >
> > (Q4) Layerwise pruning ratio.--------------------------------------------------------------------------------
> >
> > (A4) The main reason for using the fixed pruning ratio (identical to the setup in Frankle and Carbin [7] except for FCN) was to ensure fairness in comparing MP with LAP. As the reviewer points out, the performance-compression tradeoff of LAP could be improved if more careful sensitivity analyses have taken place, as LAP focuses on local structures only.
> >
> > However, we report following experimental results suggesting that such tuning is not always required if one’s aim is to improve over baselines. In Appendix G of the revised manuscript, we extend LAP and its variant using training data (as introduced in response to 1&2), to a global pruning scheme. In our experiment, LAPs with global pruning outperform corresponding MP and OBD baselines. Nevertheless, further tuning of the layerwise pruning ratio could improve the performance of LAP and its variants.
> >
> >
> > (Q5) LFP vs. LBP.-------------------------------------------------------------------------------------------------
> >
> > (A5) We are glad that the reviewer pointed this out; it is an intriguing observation indeed. We interpret the phenomenon as follows: Whenever the sparsity level is low, the importance of carefully curating the input signal is not significant due to a high redundancy in natural image signals and the corresponding features from over-parametrized models. On the other hand, the prediction based on given features is more directly affected by the layers closer to the output. This causes a relatively smaller gain by looking backward than forward in the low-sparsity regime. When the sparsity level is high, the input signal is scarce, and the relative importance of preserving the input signal is greater. We added notes on this intuition in Section 3.2.
> >
> > (Q6) Entire network as an operator block.---------------------------------------------------------------
> >
> > (A6) We thank the reviewer for proposing an interesting variant of LAP. We have implemented and tested this version under MNIST + 5-layer MLP (5 trials total). To our surprise, the whole-networks-as-a-block variant performed slightly worse than LAP. The experimental results are added in Appendix J of the revised manuscript. We suspect this is because we are ignoring the effect of all nonlinearities and bias terms, which may accumulate over a deep stack of layers.
> >
> > (Q7) Further experiments.-------------------------------------------------------------------------------------
> >
> > (A7) We have performed additional experiments on Tiny-ImageNet dataset on VGG-19, ResNet-50 and WRN-16-8, and have included the results in Section 3.3. Coherent to the previous observations on smaller datasets, the advantages of LAP over MP have been confirmed for Tiny-ImageNet dataset as well.
> >
> > We would continue to add more datasets and network architectures as time permits.
> >
> > -------------------------------------------------------------------------------------------------------------------------
> >
> > [6] Dong et al., “Learning to prune deep neural networks via layer-wise optimal brain surgeon,” NeurIPS 2017.
> > [7] Frankle and Carbin, “The lottery ticket hypothesis: finding sparse, trainable neural networks,” ICLR 2019.

---

> > > ### Comment · AnonReviewer5 · 2019-11-14
> > > **Thanks for your response!**
> > >
> > > I really appreciate the authors' effort during the rebuttal, and most of my concerns are addressed well.
> > >
> > > However, I still have some comments about the responses:
> > >
> > > - Such data-agnostic approaches have the following clear advantages.
> > > (1) I don't think the 'Ease of computation' will be a good reason for using MP/LAP rather than OBD in practice. Because, for pruning, we only care about how it speed-ups the inference time and the accuracy of the pruned network. If OBD can achieve much better performance than MP/LAP, there is no reason to use MP/LAP rather than OBD just because of OBD is expensive to compute.
> > > (2) I think the data-agnostic argument is valid, but needs more experiments to support it. Currently, all the experiments are conducted in the data dependent setting. By data dependent, I mean all the results are obtained by pruning a pre-trained network by either MP/LAP, and then fine-tune the network on the *original dataset*,  which means we still need to access the original training data. I would suggest authors to complete the experiments as done in Morcos et al., to show that if LAP can be still performant in the data-agnostic setting. This will provide strong evidence for your claimed advantages.
> > >
> > > - Evaluation with OBD
> > > Thanks for completing the OBD experiments, I am pretty satisfied with your related responses. I would suggest the authors move the results into the main paragraph. This will be a stronger baseline than MP for people working on pruning. (Clearly, OBD is much better than MP and LAP, but seems worse than LAP-act.)
> > >
> > > -  LFP vs. LBP
> > > It makes sense, thanks.
> > >
> > > - Further experiments.
> > > Thanks for including the experiments on Tiny-ImageNet, it makes the results more convincing.
> > >
> > > Thanks so much for your response, and the great efforts to address my concerns. I am pretty satisfied with it. I will increase my rating to 6.
> > > In the meantime, I strongly recommend the authors to include the data-agnostic experiments in the paper to support your argument, and also include OBD in each table instead of just for FCN on MNIST and Conv-6 on CIFAR-10.

---

> > > > ### Author Response · Authors · 2019-11-15
> > > > **Thanks for your response!**
> > > >
> > > > Dear reviewer,
> > > >
> > > > We are truly grateful for taking your time to provide additional recommendations and acknowledge our efforts until the very last day of the response/discussion phase.
> > > >
> > > > The comments on the benefits of data-agnostic pruning are completely to-the-point.
> > > > We would continue our efforts to enhance the final version with additional experiments on mask transfer and OBD.
> > > >
> > > > Thanks,
> > > > Authors.

---

### Author Response · Authors · 2019-11-15
**Summary of revisions.**

Dear reviewers,

We express our deepest gratitude for your constructive feedback and incisive comments on our manuscript.

In response to the questions and concerns you raised, we have carefully revised and enhanced the manuscript with the following additional experiments and discussions.

- Optimal brain damage and data-dependent LAP variants (Appendix F),
- Computational costs of LAP (Appendix G),
- Channel pruning with lookahead cost (Appendix H),
- Tiny-ImageNet dataset with VGG, ResNet, and WRN (Section 3.3, Appendix C),
- Global Pruning (Appendix I),
- Whole-network-LAP (Appendix J),
- Performance comparisons with MobileNet (Appendix K).

The revisions made are marked with “blue” in the revised manuscript.

We also appreciate your continued effort to provide further feedback until the very end of response/discussion phase. We will make sure to reflect the comments in the final version.

Thanks,
Authors.

---

### Decision · Program_Chairs · 2019-12-19

**Decision:**

Accept (Poster)

**Comment:**

This paper introduces a pruning criterion which is similar to magnitude-based pruning, but which accounts for the interactions between layers. The reviewers have gone through the paper carefully, and after back-and-forth with the authors, they are all satisfied with the paper and support acceptance.